# Mutations in *FAM50A* suggest that Armfield XLID syndrome is a spliceosomopathy

Yu-Ri Lee [1,20], Kamal Khan[2,3,4,5,20], Kim Armfield-Uhas[6,20], Sujata Srikanth[7,20], Nicola A. Thompson[8], Mercedes Pardo [9], Lu Yu [9], Joy W. Norris [7], Yunhui Peng [10], Karen W. Gripp [11], Kirk A. Aleck[12], Chumei Li[13], Ed Spence[14], Tae-Ik Choi [1], Soo Jeong Kwon [15], Hee-Moon Park[15], Daseuli Yu [16], Won Do Heo [16], Marie R. Mooney [2,3], Shahid M. Baig[4], Ingrid M. Wentzensen[17], Aida Telegrafi[17], Kirsty McWalter [17], Trevor Moreland [7], Chelsea Roadhouse[13], Keri Ramsey[18], Michael J. Lyons [7], Cindy Skinner[7], Emil Alexov[10], Nicholas Katsanis[2,3,19], Roger E. Stevenson [7], Jyoti S. Choudhary [9], David J. Adams [8], Cheol-Hee Kim [1,21✉], Erica E. Davis [2,3,19,21✉] & Charles E. Schwartz [7,21✉]

Intellectual disability (ID) is a heterogeneous clinical entity and includes an excess of males who harbor variants on the X-chromosome (XLID). We report rare *FAM50A* missense variants in the original Armfield XLID syndrome family localized in Xq28 and four additional unrelated males with overlapping features. Our *fam50a* knockout (KO) zebrafish model exhibits abnormal neurogenesis and craniofacial patterning, and in vivo complementation assays indicate that the patient-derived variants are hypomorphic. RNA sequencing analysis from *fam50a* KO zebrafish show dysregulation of the transcriptome, with augmented spliceosome mRNAs and depletion of transcripts involved in neurodevelopment. Zebrafish RNA-seq datasets show a preponderance of 3′ alternative splicing events in *fam50a* KO, suggesting a role in the spliceosome C complex. These data are supported with transcriptomic signatures from cell lines derived from affected individuals and FAM50A protein-protein interaction data. In sum, Armfield XLID syndrome is a spliceosomopathy associated with aberrant mRNA processing during development.

[1] Department of Biology, Chungnam National University, Daejeon, Korea. [2] Center for Human Disease Modeling, Duke University Medical Center, Durham, NC, USA. [3] Advanced Center for Translational and Genetic Medicine (ACT-GeM), Stanley Manne Children's Research Institute, Ann & Robert H. Lurie Children's Hospital of Chicago, Chicago, IL, USA. [4] Human Molecular Genetics Laboratory, Health Biotechnology Division, National Institute for Biotechnology and Genetic Engineering (NIBGE), Faisalabad, Pakistan. [5] Pakistan Institute of Engineering and Applied Sciences (PIEAS), Islamabad, Pakistan. [6] Children's Healthcare of Atlanta, Atlanta, GA, USA. [7] Greenwood Genetic Center, Greenwood, SC, USA. [8] Wellcome Sanger Institute, Hinxton, Cambridge, UK. [9] Chester Beatty Laboratories, Institute of Cancer Research, London, UK. [10] Department of Physics, Clemson University, Clemson, SC, USA. [11] Division of Medical Genetics, A. I. duPont Hospital for Children, Wilmington, DE, USA. [12] Genetics and Metabolism, Phoenix Children's Medical Group, Phoenix, AZ, USA. [13] Clinical Genetics Program, McMaster University Medical Center, Hamilton, ON, Canada. [14] Division of Pediatric Genetics and Metabolism, University of North Carolina School of Medicine, Chapel Hill, NC, USA. [15] Department of Microbiology and Molecular Biology, Chungnam National University, Daejeon, Korea. [16] Department of Biological Sciences, Korea Advanced Institute of Science and Technology, Daejeon, Korea. [17] GeneDx Inc, Gaithersburg, MD, USA. [18] Center for Rare Childhood Disorders, TGen, Phoenix, AZ, USA. [19] Department of Pediatrics, Feinberg School of Medicine, Northwestern University, Chicago, IL, USA. [20] These authors contributed equally: Yu-Ri Lee, Kamal Khan, Kim Armfield-Uhas, Sujata Srikanth. [21] These authors jointly supervised: Cheol-Hee Kim, Erica E. Davis, Charles E. Schwartz. ✉email: zebrakim@cnu.ac.kr; eridavis@luriechildrens.org; ceschwartz@ggc.org

ntellectual disability (ID) affects 1–3% of the general population[1]. Males exceed females in the ID population by 20–30%, likely due to an enrichment of genes on the X-chromosome that are required for neurodevelopment. X-linked ID (XLID) disorders, resulting from hemizygous variants, contribute significantly to the male ID population[2]. Efforts by research groups worldwide have identified 145 XLID genes contributing to 114 XLID syndromes and 63 non-syndromic XLID entities[3]. Exome sequencing has accelerated mutational analysis of the coding regions of the X-chromosome and identified 28 of the 145 XLID genes in the past decade[3]. Despite these accomplishments, more than 56 XLID syndromes and 33 non-syndromic XLID entities remain without a molecular diagnosis.

In 1999, we characterized Armfield XLID syndrome and localized the causal locus to an 8 Mb region on Xq28 using linkage analysis[4]. Affected individuals display a distinctive phenotype involving multiple systems: postnatal growth retardation; variable head circumference with a prominent forehead and dysmorphic facial features; ocular abnormalities and seizures[4]. Here, we report the causal variant that segregates with the Armfield syndrome phenotype. As part of a screen of XLID genes localized to Xq28, we identify an ultra-rare missense variant in *FAM50A* (family with sequence similarity 50 member A; known as *XAP5* or *HXC26*) in affected males and unaffected carrier females. We use GeneMatcher[5], to identify four unrelated males who have undergone whole-exome sequencing (WES), and who each bear a rare missense variant in *FAM50A*. These males display phenotypes similar to Armfield XLID syndrome.

To investigate FAM50A function, establish relevance to the Armfield XLID clinical spectrum, and test variant pathogenicity, we utilize zebrafish (*Danio rerio*). A zebrafish *fam50a* knockout (KO) recapitulates the human phenotype with abnormal development of cephalic structures. In addition, we use in vivo complementation studies to show that the missense *FAM50A* changes identified confer a partial loss of function. Transcriptomic studies of *fam50a* KO zebrafish heads enable correlation with the human phenotype and validate previous reports suggesting FAM50A to be associated with the spliceosome complex[6,7]. Transcriptomic data from lymphocyte cell lines (LCL) derived from affected males and FAM50A protein–protein interaction data further support the previous findings. We propose that aberrant spliceosome C-complex function is the molecular mechanism underpinning Armfield XLID, defining it as a spliceosomopathy.

## Results

**Clinical and genetic studies implicate *FAM50A* in XLID.** We report updated clinical information for affected siblings in family K8100 (IV-1 and IV-2; Fig. 1a and Table 1; Supplementary Note 1). The causal locus was localized to Xq28[4], and within this chromosome band, a hitherto uncharacterized gene, *FAM50A/XAP5*, was reported in which the 5′ untranslated region contained a run of GGC repeats[8]. Analysis of an affected male from K8100 along with males from other XLID families localized to Xq28 showed no expansions beyond the normal range. We performed bidirectional Sanger sequencing of the coding regions and exon–intron boundaries of five candidate genes located in Xq28 (*GDI1*, *MECP2*, *L1CAM*, *AFF2/FMR2*, *FAM50A/XAP5*) in affected males. These analyses revealed a missense change, c.764A>G, p.Asp255Gly, in *FAM50A* (GenBank [https://www.ncbi.nlm.nih.gov/nuccore/NM_004699.4]), which segregated with disease in the family (Fig. 1a). To exclude the possibility of a causal variant elsewhere in Xq28, we included an affected male from K8100 in a larger sequencing project of 718 genes located on the X-chromosome[9]. The same alteration in *FAM50A* was the only likely causal change identified. This same variant was again

identified as the sole candidate in K8100 as part of an X-exome next-generation sequencing project conducted later[10].

The p.Asp255Gly change was not present in 400 X-chromosomes from ethnically matched controls from our in-house data set, and is absent from 182,557 alleles in gnomAD (accessed April 2019; https://gnomad.broadinstitute.org/). Prediction algorithms suggested that 255Gly was likely pathogenic (Supplementary Table 1). Asp255 is embedded within a highly conserved string of amino acids (KEDLI) present in vertebrates, *D. melanogaster*, *C. elegans*, and *S. pombe* (Supplementary Fig. 1a–c). Secondary structure prediction programs proposed that Asp255Gly is located in a beta-turn of a random coil domain (Supplementary Fig. 2a). In silico protein modeling indicated that p.Asp255Gly is located in a short loop in the low confidence structural region of the model (Supplementary Fig. 2). In the wild-type (WT) structure, a hydrogen (H)-bond is formed between the Asp255 side chain and Arg180, and the variant is expected to affect the H-bonding network and alter the net charge. However, binding free energy change predictions made by five servers are inconsistent, likely due to fidelity of the modeled structure (Supplementary Table 2). In sum, genetics data coupled to variant prediction and protein modeling suggested that p.Asp255Gly is deleterious. We then sought additional individuals through GeneMatcher[5], and identified four unrelated males with variants in *FAM50A*.

The family K9648 proband (II-3) was 10 years old at the last clinical examination and he displayed global developmental delay, prominent forehead, glaucoma, and small hands and feet (Fig. 1b and Table 1; Supplementary Note 1). We performed parent-proband trio WES. Subsequent to bioinformatic filtering for rare functional variants under autosomal recessive, dominant de novo or X-linked paradigms, we identified a maternally inherited *FAM50A* c.616T>G; p.Trp206Gly variant (Fig. 1b; Supplementary Fig. 1a–c). All prediction algorithms indicated that the p.Trp206Gly variant was likely pathogenic (Supplementary Table 1). In silico protein modeling analysis indicated that p.Trp206Gly is located in a short loop of the high-confidence structural region, partially buried and surrounded by hydrophobic residues (Supplementary Fig. 2a, b). However, protein folding free energy changes in the context of Trp206Gly are contradictory (Supplementary Table 2).

The family K9656 proband (II-1) presented with global delay, strabismus, short stature, and dysmorphic facial features (Fig. 1c and Table 1; Supplementary Note 1). WES of the proband and his healthy parents identified a hemizygous, de novo, variant in *FAM50A* (c.761A>G, p.Glu254Gly; Fig. 1c; Supplementary Fig. 1a–c). The variant was predicted to be deleterious (Supplementary Table 1), and occurs in a short loop in the low confidence structural region of the protein (Supplementary Fig. 2a, b). Glu254 makes an H-bond with nearby residue Asn177 and the substitution with Gly will eliminate the H-bond. The variant is also predicted to destabilize the structure and to alter the net charge at position 254 (Supplementary Table 2).

The clinical presentation of the family K9667 affected male (II-2) consisted of global delay, exotropia, and myopia, although he did not have short stature (Fig. 1d; Table 1; Supplementary Note 1). WES of the proband and his unaffected parents identified a de novo *FAM50A* variant (c.817C>T, p.Arg273Trp; Fig. 1d; Supplementary Fig. 1a–c). All bioinformatic analyses indicated that the variant was likely pathogenic (Supplementary Table 1). Protein modeling suggests that p.Arg273Trp is located in a helix in the high-confidence structure (Supplementary Fig. 2a, b); the side chain of Arg273 is buried and forms H-bonds with Glu200 and the backbone of Ile199. This amino acid change potentially affects protein stability, and predictions from different

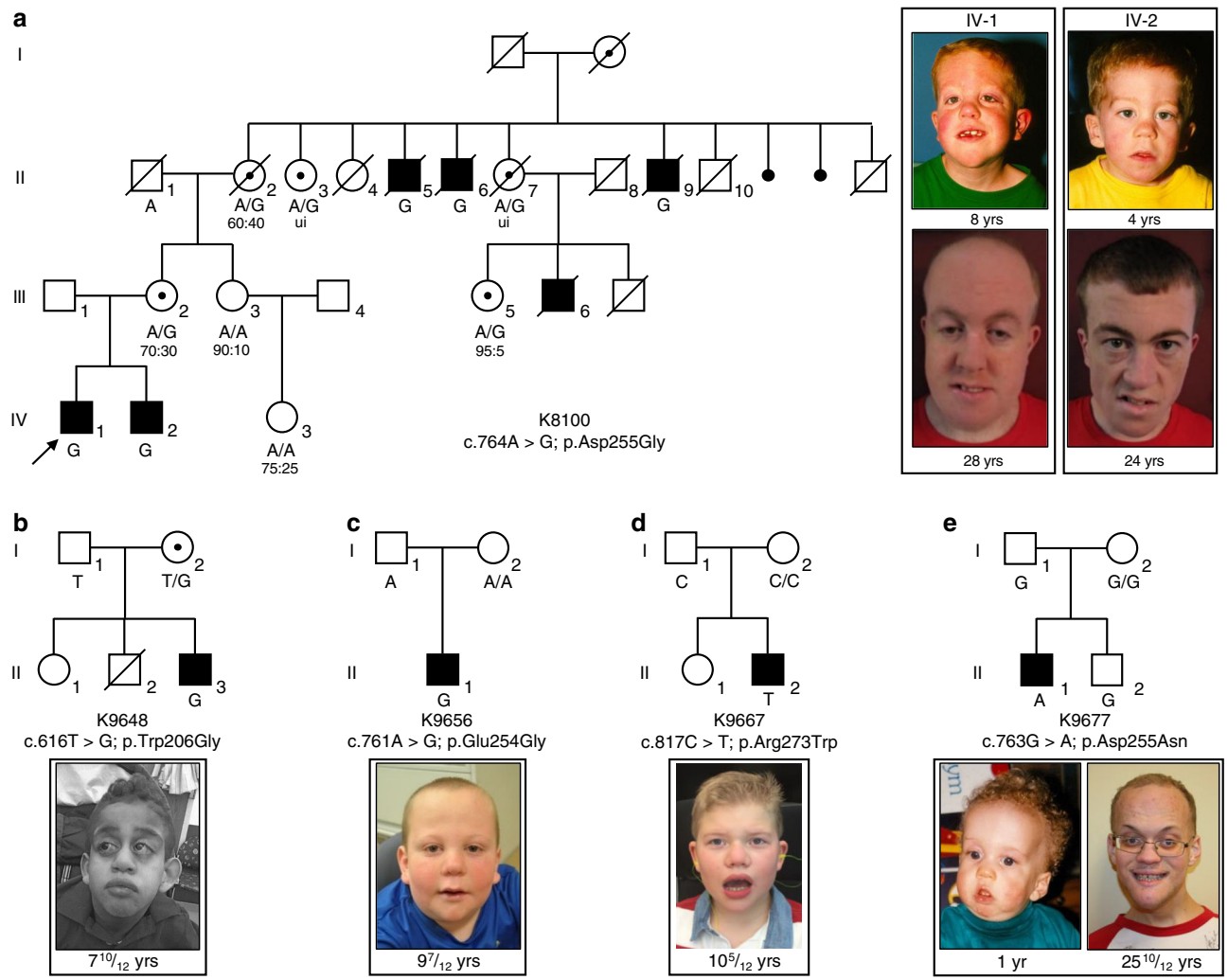

**Fig. 1 Missense variants in *FAM50A* cause XLID in five unrelated families. a–e** Pedigrees of the five families reported in this study are shown, with *FAM50A* genotype given for each available individual. Photographs of available affected males are provided for each pedigree. For family K8100, photographs are provided for the two affected males in generation IV at ages 8 and 4 years, when the family was originally published[4]; new photos from the last clinical assessment (December 2017) are shown (28 and 24 years). Ratios under females II-2, III-2, III-3, III-5, and IV-3 represent X-inactivation data. Females II-3 and II-7 were uninformative (ui) at the *AR* locus. Circles, females; squares, males; unfilled shapes, unaffected; black filled shapes, affected; unshaded circle with black dot, carrier female as determined by *FAM50A* analysis or by pedigree structure; diagonal line, deceased. Male K8100-III-6 had macrocephaly, seizure disorder, bilateral ventricular enlargement, and atrophy of the left hemisphere on a pneumoencephalogram; he was unavailable for *FAM50A* genotyping.

servers also indicate that the Arg273Trp variant might destabilize the protein (Supplementary Table 2).

The clinical features of the family K9677 male proband (II-1) overlapped with affected males in K8100 (Fig. 1e and Table 1; Supplementary Note 1); he exhibited global developmental delay, dysmorphic facial features and exotropia. Trio-based WES and bioinformatic filtering identified a rare de novo variant in *FAM50A* (c.763G>A, p.Asp255Asn; Fig. 1e; Supplementary Fig. 1a, b), that was predicted to be likely pathogenic (Supplementary Table 1). Protein modeling suggests that this change introduces a polar residue, Asn, and thus does not form an H-bond with residue Arg180 as does the WT amino acid (Supplementary Fig. 2). Protein stability predictions are inconsistent (Supplementary Table 2). However, the variant alters the charge at residue 255, potentially affecting FAM50A function.

Of the five families with *FAM50A* variants, the variants in K8100 and K9648 were inherited. All available females in K8100 were tested for X-inactivation (XI) at the *AR* locus[11]. We observed no correlation between the presence of the *FAM50A*

alteration and the degree of skewed XI (Fig. 1a). Both a noncarrier female (III-3) and a carrier female (III-5) had significant skewing of XI. The three remaining informative females (carriers II-2, III-2; non-carrier IV-3) had random-to-moderate skewed XI.

Together, we identified a cohort of nine males from five unrelated families who carry rare *FAM50A* variants. Affected individuals share syndromic ID and comorbid phenotypes impacting growth, facial gestalt, and ocular development (Fig. 1 and Table 1). These nonsynonymous changes segregate with disease status in pedigrees, are absent from gnomAD, and reside within highly conserved regions of the XAP domain in the C-terminal portion of FAM50A (Fig. 1; Supplementary Fig. 1a, b and Supplementary Table 1).

**FAM50A has ubiquitous expression and nuclear localization.** The *FAM50A* cDNA was characterized through efforts to catalog the genes on Xq28[8,12]. Monitoring of *FAM50A* in adult and fetal human tissue panels reported ubiquitous expression[8,13]. We

**Table 1 Clinical findings in five families with mutations in *FAM50A*.**

| Individual | K8100 IV-1 | K8100 IV-2 | K9648 II-3 | K9656 II-1 | K9667 II-2 | K9677 II-1 |
|---|---|---|---|---|---|---|
| *FAM50A* variant | c.764A>G; p.Asp255Gly | c.764A>G; p.Asp255Gly | c.616T>G; p.Trp206Gly | c.761A>G; p.Glu254Gly (de novo) | c.817C>T; p. Arg273Trp (de novo) | c.763G>A; p. Asp255Asn (de novo) |
| Ethnicity | Caucasian | Caucasian | Mixed (African-American, Middle Eastern, Mixed European) | Caucasian | Caucasian | Caucasian |
| **Clinical characteristics** | | | | | | |
| **Growth** | | | | | | |
| Birth (gestational weeks) | 40 | 40 | 34 | 35 | 38.5 | ND |
| Length, cm (%) | 47 (5) | 45.7 (<3) | 43.2 (15) | 47 (50) | 48.3 (25) | 53.3 (95) |
| Weight, kg (%) | 3.2 (30) | 2.9 (15) | 2.4 (60) | 2.5 (40) | 2.8 (20) | 4.4 (97) |
| HC, cm (%) | 36.2 (55) | 36 (50) | NA | NA | 34.3 (20) | 39 (>97) |
| Postnatal (years–months) | 28 | 24 | 7–10 | 9–7 | 8–3 | 25–10 |
| Height, cm (%) | 154.9 (<3) | 154.9 (<3) | 106.2 (<3) | 120.5 (<3) | 122 (9) | 160.5 (<3) |
| Weight, kg (%) | 70 (15) | 43.2 (<3) | 19.4 (<3) | 38.4 (85) | 27.9 (63) | 61.9 (<3) |
| HC, cm (%) | 58.6 (85) | 58.5 (85) | 50.2 (5) | 54 (75) | 52.1 (30) | 60.25 (>97) |
| **Development** | | | | | | |
| Delay | Global, special education, ambulatory, speaks in short phrases | Global, special education, ambulatory (walked at 3 yrs) single words | Global, not ambulatory, no speech | Global, regular classes with support | Global, no speech, special education, ambulatory for short distances | Global |
| IQ | 66 | ND | ND | ND | <50 | 63 |
| **Somatic findings** | | | | | | |
| Craniofacial | Macrocephaly, epicanthal folds, depressed nasal bridge, downslanted palpebral fissures, cleft palate, bow-shaped mouth, microretrognathia | Broad forehead, epicanthal folds, depressed nasal bridge, downslanted palpebral fissures, low-set ears, micrognathia | Prominent forehead, bitemporal narrowing, proptosis, hypotelorism, tubular nose, single median incisor, hypodontia, low-set ears, large left ear, prominent lips | Bilateral epicanthic folds, infraorbital creases, wide nasal root, short and lightly upturned nose with underdeveloped nares, slightly posteriorly rotated ears, faint hemangiomas between brows and at back of neck | Bulbous nose, excessively folded helices | Prominent tall forehead, overfolded helices, micrognathia |
| Ocular | Strabismus | — | Axenfeld-Rieger with glaucoma, nystagmus | Strabismus | Exotropia | Exotropia, keratoconus, nystagmus |
| Cardiac | ASD, PDA | — | Tetralogy of Fallot, right ventricle dilation | ASD | — | |
| Skeletal | Small feet, pes cavus, hammertoes | Small feet, pes cavus, hammertoes | Small hands/feet, short limbs, crease across dorsum of feet, foot eversion/inversion, coxa valga, mild scoliosis | Joint hypermobility | Small hands/feet | Stiff joints, small hands and feet, club foot |
| Gastrointestinal | — | — | G-tube | Inguinal hernia | Hiatal hernia, dysphagia, constipation | Umbilical hernia, imperforate anus |
| Genitourinary | — | — | Horseshoe kidney, micropenis, undescended testis | Unilateral renal agenesis, micropenis, small scrotum | Cryptorchidism | — |
| Skin | Capillary hemangiomas of nasal bridge and eyelids | Facial capillary hemangiomas | Sacral dimple, lipoma | ND | ND | Hemangiomas |
| Neurologic | Seizures | Seizures | Hypotonia | Hypotonia | Hypotonia/hypertonia, jerky movements, tethered cord | Seizures, tremor, hypotonia, incontinent |
| Other | Obstructive sleep apnea | Aggressive, quick tempered | Sleep disturbance | 2 vessel umbilical cord, obesity, gynecomastia | Sleep disturbance, incontinent | Impulsive, mood disorder, hypothyroidism, hypodontia |
| MRI | Enlarged 3rd ventricle, extra-axial fluid | Asymmetric ventricles | Decreased white matter, small corpus callosum, small brain stem | — | — | — |

ASD, atrial septal defect; G-tube, gastrostomy tube; HC, head circumference; IQ, intelligence quotient; MRI, magnetic resonance imaging; ND, no data; PDA, patent ductus arteriosus; yrs, years; (—) not present.

performed semi-quantitative RT–PCR using a multiple human fetal tissue cDNA panel (Clontech). *FAM50A* was expressed in all eight fetal tissues assessed, including brain (Supplementary Fig. 3a). Next, we evaluated *FAM50A* expression in fetal brain using a Rapid Scan Human Brain Panel (Origene). *FAM50A* was detectable in the fetal cerebellum and hypothalamus (Supplementary Fig. 3b). However, we observed low expression of *FAM50A* in the temporal lobe and were unable to detect expression in the hippocampus using this method.

Mazzarella and colleagues[8] reported features suggestive of nuclear localization in the FAM50A amino acid sequence. cNLS mapper (http://nls-mapper.iab.keio.ac.jp/cgi-bin/NLS_Mapper_form.cgi) indicated that FAM50A (GenBank ID: NP_004690.1) contained a nuclear localization signal (NLS), ITTKKRKLG (positions 149-157, score of 7.5), predicting partial localization to the nucleus[14]. cNLStradamus (http://www.moseslab.csb.utoronto.ca/NLStradamus), predicted a NLS domain within FAM50A amino acids 88–116 (LAKKEQSKELQMKLEKLREKERK-KEAKRK)[15]. To examine FAM50A cellular localization, we performed endogenous protein immunostaining of NIH/3T3 cells. We observed dispersed nuclear localization throughout the cell cycle (Supplementary Fig. 4a). After demonstrating that FAM50A protein levels are not significantly different in LCLs derived from affected males vs matched controls (Supplementary Fig. 5), we tested whether the XLID-associated variants affected FAM50A localization. We generated C-terminally tagged V5 plasmids containing the WT or mutant *FAM50A* open reading frame (ORF) and visualized FAM50A in transfected COS-7 cells (Supplementary Table 3). All mutant proteins tested (p. Trp206Gly, p.Glu254Gly, and p.Asp255Gly) localized to the nucleus and were indistinguishable from WT (Supplementary Fig. 4b), an unsurprising result since none of the variants impacted a predicted NLS. However, the nuclear staining in the NIH/3T3 cells appeared more diffuse than the transfected COS-7 cells, possibly reflecting the different technologies and cell types utilized.

**_fam50a_ KO zebrafish display patient-relevant phenotypes**. The zebrafish genome harbors a single reciprocal ortholog encoding a protein with the same length as the human protein (339 amino acids), which is highly conserved (86% identity; 93% similarity; Supplementary Fig. 1c). To characterize the spatiotemporal expression of *D. rerio fam50a*, we probed RNA in situ on whole-mount embryos at eight different stages. We noted ubiquitous expression prior to and throughout gastrulation, and in mid-somitic embryos. From 24 h post-fertilization (hpf) onward, and up to the latest time point assessed (72 hpf) *fam50a* mRNA regionalized to all visible anterior structures including the brain, eye and mandible (Supplementary Fig. 6a). These observations were consistent with publicly available zebrafish RNA-seq data (https://www.ebi.ac.uk/gxa/experiments/E-ERAD-475/Results).

To determine the consequences of FAM50A depletion, we performed genome editing and identified loss of function mutant alleles. We disrupted the highly conserved XAP domain by injecting Cas9 mRNA and guide RNA targeting either *fam50a* exon 6 or exon 7, screened for mosaic F0 founders, and isolated two mutant alleles in the F1 generation (compound 11 + 5 bp deletion [KO1] or 4 bp deletion [KO2]; Supplementary Fig. 6b–e). We used KO1 for all subsequent phenotyping (hereafter referred to as KO). Whole-mount in situ hybridization (WISH), immunofluorescent antibody staining, or immunoblotting of embryo-derived protein lysate could not detect *fam50a* mRNA or FAM50A protein as early as 24 hpf (Supplementary Fig. 6f, g, h). *fam50a* KO was present in the expected Mendelian ratios and displayed similar gross morphology to WT until ~3 days post-

fertilization (dpf). However, at 5 dpf KO larvae were severely affected by abnormal anterior development impacting the brain, eyes and cartilage (Fig. 2a), resulting in lethality by ~6 dpf.

We evaluated cell-type specific and cellular response assays in *fam50a* KO and WT larvae (Supplementary Table 3). Using a transgenic reporter of pan-neuronal cells, *tg(huc:egfp)*, we found a reduction of differentiated neurons in *fam50a* KO brain (Fig. 2b), consistent with *her4.1* WISH indicating diminished neurogenesis[16] at 3 dpf (Fig. 2c; Supplementary Fig. 7). Both markers showed indistinguishable neuronal integrity at 2 dpf (Fig. 2b, c). We tested whether altered proliferation and cellular stress responses could account for the onset of neuronal phenotypes. We observed marked depletion of proliferation markers *pcna* and *ccnd1* at 3 dpf with concomitant augmentation of p53 pathway effectors *tp53*, *mdm2*, and *cdkn1a* at an earlier stage, especially in the midbrain (2 dpf; Fig. 2d; Supplementary Fig. 8). Blood vessel development was normal up to 2.5 dpf in *tg(kdrl:egfp);fam50a* KO zebrafish (Supplementary Fig. 9a), and was confirmed by in situ analysis with endothelial molecular markers *etv2*, *cdh5*, and *cldn5b* (Supplementary Fig. 9b). These data recapitulate specific early neurogenesis defects that are present in Armfield XLID syndrome.

Affected males with mutations in *FAM50A* show dysmorphic facial features (Fig. 1 and Table 1). We stained zebrafish larvae with Alcian blue at different time points (2.5, 3, and 4.5 dpf) to study orthologous structures. At 2.5 dpf, we observed no major differences between KO and WT siblings; cartilage structures such as the ceratohyal, palatoquadrate, and ethmoid plate were present. However, by 3 dpf, we observed anterior-posterior shortening of the pharyngeal skeleton with delayed branchial arch patterning. This persisted up to 4.5 dpf (Fig. 2e). To quantify these defects, we generated −1.4col1a1:egfp;fam50a larvae and performed live ventral imaging of fluorescent signal at 3 dpf and measured the ceratohyal angle as a proxy for mandibular development. We observed a significantly wider ceratohyal angle in *fam50a* KO compared to heterozygotes or WT ($p = 5.3E{-}13$: unpaired Student's *t*-test, two-sided; repeated; Fig. 3a, b; Supplementary Fig. 10a and Supplementary Table 4).

**Zebrafish studies show that _FAM50A_ variants are hypomorphic**. Cross-species in vivo complementation testing is a sensitive and specific method to establish missense variant pathogenicity[17]. We targeted the splice donor site of *D. rerio fam50a* exon 4 with a morpholino (MO), and RT–PCR showed deletion of exons 3 and 4, and a subsequent frameshift and premature transcript termination (Supplementary Fig. 11a–c). We injected increasing doses of *fam50a* splice-blocking MO (3, 6, and 9 ng) into −1.4col1a1:egfp embryos and assessed ventral cartilage at 3 dpf using the live automated imaging paradigm. We observed a dose dependent exacerbation of craniofacial features ($p = 0.34$, $1.5E{-}38$, and $4.3E{-}31$ for 3, 6, and 9 ng vs control, respectively; unpaired Student's *t*-test, two-sided, repeated; Supplementary Fig. 11d and Supplementary Table 4), and cartilage-patterning defects matched the *fam50a* KO (Figs. 2e and 3a; Supplementary Fig. 10a). We rescued this phenotype by co-injecting either 150 pg of human *FAM50A* WT mRNA or mRNAs carrying common variants (gnomAD) as negative controls (p.Ala137Val and p.Glu143Lys; $p = 2.4E{-}12$, $9.9E{-}13$, and $5.7E{-}11$ for WT or variant rescue vs MO, respectively; unpaired Student's *t*-test, two-sided, repeated; Figs. 3a, c; Supplementary Fig. 10b, Supplementary Tables 3 and 4).

Next, we tested the patient-specific allelic series (p.Trp206Gly, p.Glu254Gly, p.Asp255Gly, p.Asp255Asn, and p.Arg273Trp) using equivalent doses of MO and mRNA across experiments (Supplementary Tables 3 and 4). Co-injection of MO with all

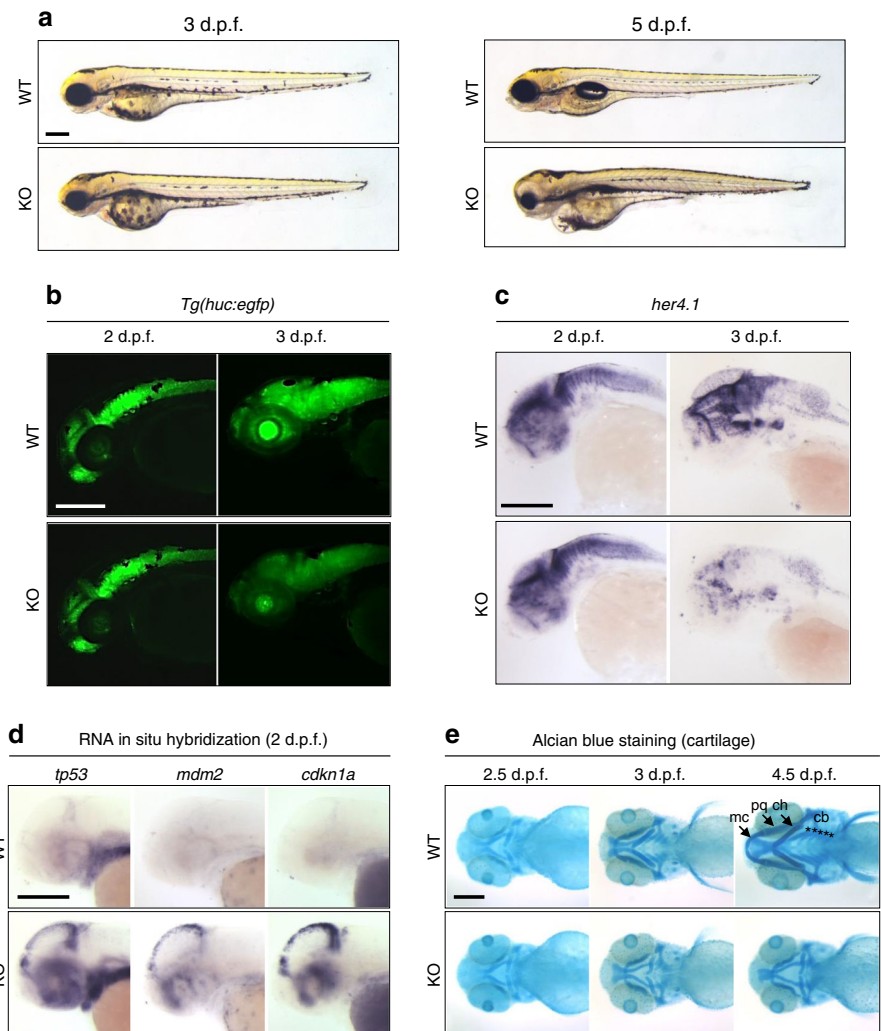

**Fig. 2 fam50a KO zebrafish display central nervous system and craniofacial patterning defects. a** Representative bright field lateral images of WT and *fam50a* KO are shown at 3 and 5 days post-fertilization (dpf). Morphology of *fam50a* KO was relatively normal at 3 dpf; repeated four times. However, at 5 dpf, *fam50a* KO showed craniofacial abnormalities; repeated five times. Number of larvae assessed with similar results: 3 dpf, $n = 56$; 5 dpf, $n = 44$. **b** Fluorescent lateral images of WT and *fam50a* KO larvae on a *Tg(huc:egfp)* neuronal reporter. No difference between WT and *fam50a* KO was detected in the anterior structures at 2 dpf. However, at 3 dpf, a prominent reduction of GFP-positive neurons was observed in KO larvae. Number of larvae assessed with similar results: 2 dpf, $n = 20$; 3 dpf, $n = 20$. **c** Whole-mount in situ hybridization (WISH) of *her4.1*, a molecular marker of neurogenesis, indicated a depletion of neurons in *fam50a* KO compared to WT at 3 dpf; repeated. Number of larvae assessed with similar results: 2 dpf, $n = 24$; 3 dpf, $n = 21$. **d** Apoptosis markers are elevated in *fam50a* KO larvae at 2 dpf. Note the induction of *tp53* and *tp53* target genes, *mdm2* and *cdkn1a* (p21) in the cell proliferative zone in the midbrain region *fam50a* KO larvae at 2 dpf; repeated. Number of larvae assessed with similar results: *tp53*, $n = 41$; *mdm2*, $n = 52$; *cdkn1a*, $n = 39$. **e** Representative ventral images of Alcian blue staining of cartilage structures shows severe defects in cartilage development that become apparent at 3 dpf. Meckel's cartilage, mc; palatoquadrate, pq; ceratohyal arch, ch; and ceratobranchial arches, cb. Number of larvae assessed with similar results: 2.5 dpf, $n = 28$; 3 dpf, $n = 21$; 4.5 dpf, $n = 14$. For all images: anterior, left; posterior, right. Scale bars; 200 μm.

patient variants resulted in a mean ceratohyal angle significantly broader than *FAM50A* WT ($p = 0.0025$, 0.0003, 3.3E−13,1.06E−09, and 0.0004, respectively, unpaired Student's *t*-test, two-sided, repeated; Fig. 3c; Supplementary Fig. 10b–e). The failure of mutant mRNA to rescue morphant phenotype was unlikely due to dominant toxic effects; expression of *FAM50A* mRNAs alone did not result in significant defects (replicated; Supplementary Fig. 12; Supplementary Table 4). We performed similar rescue experiments using *fam50a* KO zebrafish and the formation of the swim bladder as a qualitative criterion for morphology[18]. KO phenotypes were rescued by WT *FAM50A*, but were only partially improved by a subset of three *FAM50A* patient variants (Supplementary Fig. 13). In vivo complementation data, generated in either transient or stable *fam50a* models, supported a partial loss of FAM50A function in Armfield XLID syndrome.

**fam50a KO zebrafish have altered expression profiles**. We hypothesized that impaired FAM50A function might affect transcriptional regulation or mRNA processing. These predictions are supported by affinity-purified complexes that suggested FAM50A as a potential spliceosome protein[7]; and an in vitro study that classified FAM50A as a candidate mRNA binding protein[6]. We performed RNA-seq analyses on total RNA obtained from WT and *fam50a* KO larvae harvested at 2 dpf, prior to the onset of major morphological defects (Fig. 2; Supplementary Fig. 8). We obtained embryos from five pairs of $fam50a^{+/-}$ (heterozygous) adults; decapitated larvae for RNA extraction (heads) and genotyping (tails); and combined 20 genotype-matched heads per pool ($n = 5$ biological replicates of sibling-matched WT and KO; Fig. 4a). We generated ~37 million 50 bp single-read sequences per library and assessed global transcriptomic profiles in KO vs WT.

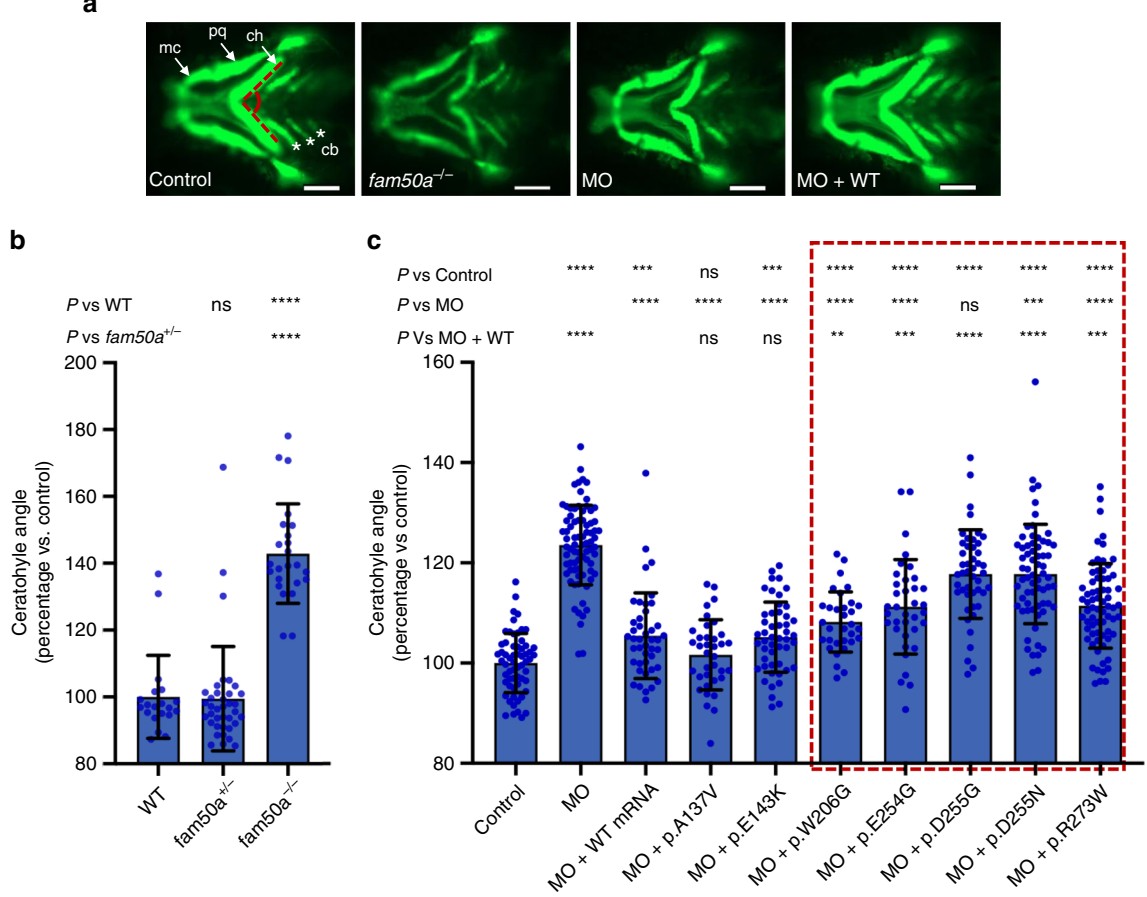

**Fig. 3 In vivo assays indicate that *FAM50A* missense variants confer a partial loss of function. a** Representative ventral views of craniofacial structures imaged live using the VAST BioImager in -1.4col1a1:egfp zebrafish larvae at 3 days post-fertilization (dpf). mc, Meckel's cartilage; pq, palatoquadrate; ch, ceratohyal arch; cb, ceratobranchial arches. Red dashed lines on control image indicate the ch angle measured to quantify altered cartilage patterning in wild-type (WT) control, KO (homozygous mutants) or morphants. 5 ng morpholino (MO) and/or 150 pg of human *FAM50A* mRNA were injected for all assays. Scale bars, 100 μm. **b** Quantification of ch angle in *fam50a* KO, *fam50a*[+/−] (heterozygous mutant) and WT. Heterozygous and WT animals are indistinguishable; ch angle was significantly increased in *fam50a* KO. **** indicates $p < 0.0001$. ns, not significant (unpaired Student's *t*-test, two-sided). See Supplementary Table 4 for exact *p*-values. Left to right: $n = 20$, 37, and 24 larvae per condition, respectively. **c** In vivo complementation studies indicate that *FAM50A* variants in XLID males are pathogenic. Quantification of ch angle as indicated by measurements of ventral images (**a**), and statistical comparison of variant mRNA vs WT mRNA rescue of MO effect indicates that patient-associated variants are hypomorphic (partial loss of function; red dashed box). p.Ala137Val (A136V; rs149558328) and p.Glu143Lys (E143K; rs782017549) are present in hemizygous males in gnomAD and were scored as benign; *, **, ***, **** indicate $p < 0.05$; 0.01; 0.001; and 0.0001, respectively. ns, not significant (unpaired Student's *t*-test, two-sided). Left to right: $n = 61$, 78, 43, 36, 49, 32, 36, 46, 69 larvae per condition, respectively; replicated. Data are presented as mean values ± standard deviation. See Supplementary Table 4 for exact *p*-values.

As expected, *fam50a* was the most significantly reduced coding mRNA between the two genotypic groups. Clustering analysis suggested a marked effect of genotype on global transcription (Supplementary Fig. 14). This observation was supported by gene-level expression analysis: ~12% of genes had significantly altered levels in *fam50a* KO compared to WT ($n = 2804$ genes, $p < 0.05$, Wald test, FDR-corrected using the Benjamini–Hochberg method), of which ~48% were downregulated ($n = 1359$ genes) and ~52% were upregulated ($n = 1445$ genes; Fig. 4b; Supplementary Fig. 15). To test whether dysregulated genes in *fam50a* KO have been implicated in human pathologies overlapping Armfield XLID, we overlaid our RNA-seq data with Human Phenotype Ontology (HPO, https://hpo.jax.org/) and Online Mendelian Inheritance in Man (OMIM, https://omim.org/) annotations. Among genes with significantly altered gene expression in *fam50a* KO, some cause clinically similar genetic disorders. Examples include downregulated genes *gss* (logFC −1.4; $p < 4E{-}26$; Glutathione synthetase deficiency) and *aaas* (logFC −1.2; $p < 1.8E{-}12$; Achalasia-addisonianism-alacrimia syndrome) and upregulated genes *ggt1b* (logFC 2.9; $p < 3.7E{-}06$; ID); and *eftud2* (logFC 1.5; $p < 1E{-}142$; Mandibulofacial dysostosis; Wald test, FDR-corrected using the Benjamini–Hochberg method; Table 2). Although a substantial fraction of the transcriptome is differentially expressed in the context of FAM50A loss, we could not identify a single altered gene driver of Armfield XLID syndrome.

Next, we performed gene set enrichment analysis (GSEA) on WT and *fam50a* KO RNA-seq data sets. The top ten pathways with significant downregulation (familywise-error rate [FWER] $p < 0.03$, Kolmogorov–Smirnov test) can be mapped to patient phenotypes, especially neurodevelopment, brain function, and cartilage patterning (Fig. 4c). However, for the top ten ranking gene sets with significant upregulation (FWER $p < 0.008$, Kolmogorov–Smirnov test), we observed that nine out of ten are involved in mRNA processing or splicing (Fig. 4c). These data suggest that FAM50A impairment leads to cellular compensation

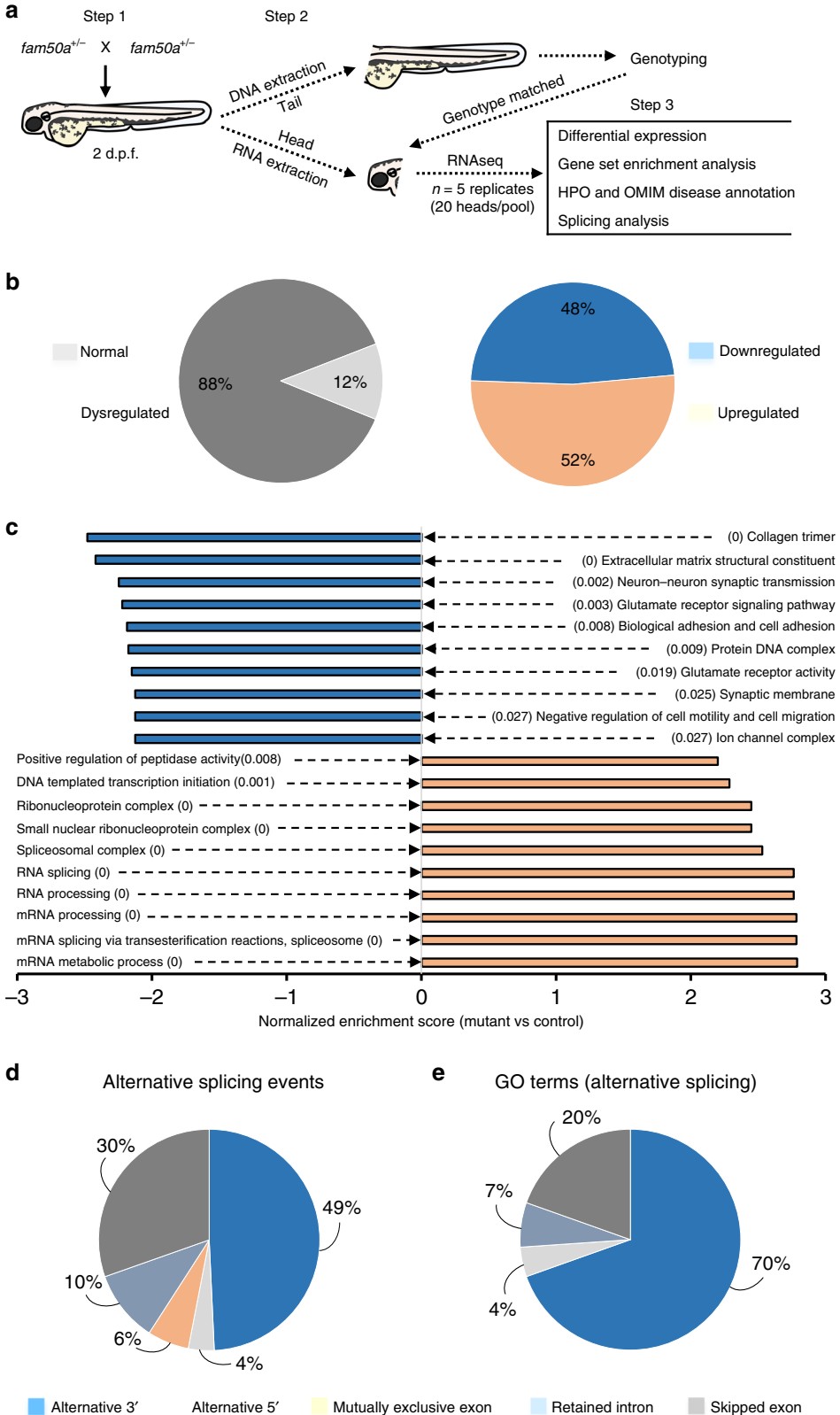

of mRNA processing effectors, arguing that neuronal and cartilage-related biological processes are particularly susceptible to FAM50A loss of function.

To validate a subset of RNA-seq results, we probed the transcript levels of 25 genes using WISH at 2 and 3 dpf (Fig. 5 and Table 2; Supplementary Fig. 16 and Supplementary Table 3).

We prioritized the top eight significantly upregulated genes (*snapc4, ice1, tp53, mdm2, mettl16, prpf3, prpf31,* and *eftud2*), the majority of which comprise the splicing machinery. We validated additional mRNA splicing effectors (*n* = 7: *prpf8, prpf4, prpf6, snrnp200, snrpe, sf3b4,* and *eif4a3*)[19–25]. Many of these genes involved in spliceosomal function are related to human disorders,

**Fig. 4 RNA-seq analysis revealed mRNA splicing defects in *fam50a* KO zebrafish. a** Schematic describing the RNA-seq experiment from sample preparation to data analysis. Steps 1 and 2 describe sample collection and preparation, whereas step 3 indicates the RNA data analysis and interpretation. Each replicate pool ($n = 5$ per genotype) contained total RNA from 20 genotype-matched larval heads at 2 days post-fertilization (dpf). *fam50a*$^{+/-}$ (heterozygous mutant). **b** Pie charts representing differential expression analysis results for KO vs WT. Transcripts with FDR-corrected $p < 0.05$ were included (Wald test, FDR-corrected with the Benjamini–Hochberg method). **c** Gene set enrichment analysis was performed using normalized enrichment score for KO vs WT. The top ten significantly disrupted pathways (depleted or augmented) are plotted along the *x* axis. Downregulated gene sets, orange; upregulated gene sets, blue. *p*-values (FDR-corrected) are indicated in parentheses (Kolmogorov–Smirnov test). **d** Pie chart representing percentage of alternative splicing events (by category) that are enriched in *fam50a* KO. $p < 0.05$, likelihood-ratio test with FDR correction. **e** Pie chart representing the distribution of GO terms impacted by discrete alternative splicing events (by category) in *fam50a* KO. $p < 0.05$, Fisher's exact test.

especially those with craniofacial or ocular phenotypes[26,27]. We also validated five transcripts that are significantly depleted in the *fam50a* KO (*gss, aaas, vwa7*, and *eda*); and one transcript involved in XLID (*huwe1*)[28] but not significantly altered in *fam50a* KO. For all 18 genes, the RNA ISH data confirmed our transcriptomic profiling (Table 2 and Fig. 5; Supplementary Fig. 16). In addition, we queried the RNA-seq data for genes assessed via WISH during the initial characterization of the *fam50a* KO. Accordingly, p53 pathway markers *tp53, cdkn1a* and *mdm2* showed significantly augmented expression ($p < 9.8E−103$, Wald test, FDR-corrected using the Benjamini–Hochberg method; $\geq$2-fold increase; Fig. 2d). Further, *her4.1, pcna*, and *ccnd1* WISH showed no significant differences, consistent with the spatiotemporal expression studies at 48 hpf (Fig. 2c and Table 2; Supplementary Figs. 7 and 8).

***fam50a* KO are enriched for mRNA mis-splicing events**. To probe the involvement of FAM50A in mRNA splicing, we applied replicate multivariate analysis of transcript splicing (rMATS), a statistical method that can detect: alternative 5′ splice sites; alternative 3′ splice sites; retained introns; mutually exclusive exons; or skipped exons from multiple replicate samples[29]. We queried splice sites and found representation of significantly augmented or depleted splicing events in each of the five categories. However, we noticed an uneven distribution of aberrant splicing events by splicing category that withstood an FDR-corrected $p < 0.05$ threshold (likelihood-ratio test; Fig. 4d). Aberrant 3′ splicing was predominantly affected ($n = 235$, 49%), followed by exon skipping ($n = 145$, 30%), retained intron ($n = 50$, 10%), mutually exclusive exons ($n = 29$, 6%), and alternative 5′ splicing ($n = 18$, 4%).

We applied the DAVID functional annotation clustering tool (https://david.ncifcrf.gov/) to the rMATS output to identify enrichment of transcript modules impacted significantly by aberrant splicing. The majority of significant gene category hits (uncorrected $p < 0.05$, Fisher's exact test) also impacted alternative 3′ splice sites ($n = 32$, 70%; Fig. 4e; Supplementary Table 5). There were only six Bonferroni-corrected ($p < 0.05$, Fisher's exact test) gene ontology (GO, http://geneontology.org/) terms cumulatively spanning all splicing categories. These GO groups were either impacted in both alternative 3′ splicing and exon skipping: RNA binding (GO:0003723); or were exclusive to alternative 3′ splicing events: nucleus (GO:0005634); DNA binding (GO:0003677); regulation of transcription, DNA-templated (GO:0006355); nucleic acid binding (GO:0003676); and transcription factor complex (GO:0005667). In sum, mRNA splicing analysis uncovered aberrant events biased toward 3′ alternative splicing or exon skipping. These alterations occur late in mRNA splicing, consistent with phenomena expected to be downstream of a C-complex impairment[30].

Transcriptomic profiling of a zebrafish *eftud2* mutant model demonstrated global RNA splicing deficiency with concomitant p53-dependent apoptosis[31], concordant with our RNA ISH and RNA-seq data. *tp53* was significantly upregulated in *fam50a* KO,

similar to its downstream target *mdm2*, compared to WT (Table 2). To test whether *p53* pathway activation is correlated with apoptosis, we examined *fam50a* morphants via TUNEL staining (Supplementary Table 4). TUNEL positive cells were significantly augmented in an anterior region of interest (ROI) ($p = 4.8E−06$, unpaired Student's *t*-test, two-sided). This defect was rescued by co-injection of human WT *FAM50A* mRNA ($p = 1.5E−06$, unpaired Student's *t*-test, two-sided; Supplementary Fig. 17a, b). Cell-cycle progression was also increased in *fam50a* morphants as determined by significantly increased phospho-histone H3 immunostaining, a marker of G2/M transition ($p = 1.5E−07$; unpaired Student's *t*-test, two-sided; Supplementary Fig. 17c, d; Supplementary Table 4). To examine whether the p53 pathway is a cause or effect of the *fam50a* KO phenotype, we crossed *fam50a*$^{+/-}$ and *tp53*$^{-/-}$ (homozygous) mutant lines. *fam50a* KO and *fam50a/tp53* double KO displayed similar phenotypes at 5 dpf (Supplementary Fig. 18). These data reinforce the involvement of p53 independent apoptosis in the *fam50a* KO phenotype that is likely correlated, directly or indirectly, with aberrant mRNA splicing.

**Patient LCLs have mRNA expression and splicing defects**. To test for transcriptional dysregulation and spliceosome disruption in LCLs derived from patients with Armfield XLID, we performed RNA-seq (K9648, p.Trp206Gly and K9656, p.Glu254Gly: Fig. 1b, c and Table 1). Differential expression analysis of triplicates showed significant dysregulation in a substantial fraction of the transcriptome in affected individuals compared to control (~38%; FDR-corrected $p < 0.05$, Wald test, FDR-corrected using the Benjamini–Hochberg method). Mapping of dysregulated transcripts in human to zebrafish RNA-seq data showed that 42% of upregulated and 39% of downregulated human transcripts, respectively, were also significantly altered in the *fam50a* KO. GSEA identified the spliceosome as the fourth most significantly enriched functional group (FWER $p < 0.0001$, Kolmogorov–Smirnov test; Supplementary Fig. 19). Similar to *fam50a* KO, we noticed an uneven distribution of aberrant splicing events by category that withstood an FDR-corrected $p < 0.05$ threshold (likelihood-ratio test). Exon skipping was predominantly affected ($n = 443$, 53%), followed by retained intron ($n = 172$, 20%), alternative 3′ splicing ($n = 136$, 16%), alternative 5′ splicing ($n = 75$, 9%), and mutually exclusive exons ($n = 15$, 2%). We performed DAVID analysis using rMATS data, however none of the Bonferroni-corrected functional groups identified in LCLs could be mapped back to the gene sets identified in the zebrafish mutant rMATS analysis (Fisher's exact test, Supplementary Table 6). These data offer consistency in upregulation of the spliceosome, with modest differences in alternative splicing events possibly because lymphocytes are not known to be a site of pathology in Armfield XLID syndrome.

**FAM50A interacts with spliceosome U5 and C-complex proteins**. To validate a potential affiliation for FAM50A in the

**Table 2 Validation of RNA-seq data using whole-mount in situ hybridization in zebrafish.**

| Gene name | Description | Human phenotype | RNA-seq log$_2$(FC) | RNA-seq p-value | In situ hybridization |
|---|---|---|---|---|---|
| fam50a | Family with sequence similarity 50, member A | ND | −3 | 3.47E-233 | Downregulated |
| gss | Glutathione synthetase | Glutathione synthetase deficiency | −1.39 | 4.08E-26 | Downregulated |
| aaas | Achalasia, adrenocortical insufficiency, alacrimia | Achalasia-addisonianism-alacrimia syndrome | −1.21 | 1.83E-12 | Downregulated |
| huwe1 | HECT, UBA and WWE domain containing 1 | Mental retardation, X-linked syndromic, Turner type | −0.14 | 0.71 | Downregulated |
| vwa7 | von Willebrand factor A domain containing 7 | ND | −0.99 | 1.07E-49 | Downregulated |
| eda | Ectodysplasin A | Ectodermal dysplasia, X-linked recessive; tooth agenesis | −0.60 | 0.02 | Downregulated |
| pcna | Proliferating cell nuclear antigen | Ataxia-telangiectasia-like disorder 2 | −0.08 | 0.72 | Downregulated |
| her4.1 | Hairy-related 4, tandem duplicate 1 | ND | −0.07 | 0.9 | Downregulated |
| ccnd1 | Cyclin D1 | ND | −0.01 | 0.97 | Downregulated |
| snapc4 | Small nuclear RNA activating complex, polypeptide 4 | ND | 2.79 | 9.16E-281 | Upregulated |
| cdkn1a | Cyclin-dependent kinase inhibitor 1A | ND | 2.40 | 9.87E-103 | Upregulated |
| ice1 | KIAA0947-like (H. sapiens) | ND | 2.29 | 9.81E-220 | Upregulated |
| mettl16 | Methyltransferase-like 16 | ND | 2.23 | 4.93E-156 | Upregulated |
| tp53 | Tumor protein p53 | ND | 2.16 | 3.14E-217 | Upregulated |
| mdm2 | MDM2 oncogene, E3 ubiquitin protein ligase | ND | 1.98 | 3.88E-163 | Upregulated |
| prpf31 | PRP31 pre-mRNA processing factor 31 homolog (yeast) | Retinitis pigmentosa | 1.75 | 1.25E-147 | Upregulated |
| prpf3 | PRP3 pre-mRNA processing factor 3 homolog (yeast) | Retinitis pigmentosa | 1.7 | 5.23E-149 | Upregulated |
| eftud2 | Elongation factor Tu GTP binding domain containing 2 | Mandibulofacial dysostosis | 1.45 | 1.01E-142 | Upregulated |
| prpf4 | PRP4 pre-mRNA processing factor 4 homolog (yeast) | Retinitis pigmentosa | 1.38 | 3.95E-74 | Upregulated |
| snrpe | Small nuclear ribonucleoprotein polypeptide E | Hypotrichosis | 1.30 | 1.16E-15 | Upregulated |
| snrnp200 | Small nuclear ribonucleoprotein 200 (U5) | NA | 1.09 | 5.49E-63 | Upregulated |
| prpf8 | Pre-mRNA processing factor 8 | Retinitis pigmentosa | 0.87 | 2.46E-33 | Upregulated |
| prpf6 | PRP6 pre-mRNA processing factor 6 homolog (S. cerevisiae) | Retinitis pigmentosa | 0.72 | 6.35E-49 | Upregulated |
| eif4a3 | Eukaryotic translation initiation factor 4A3 | Robin sequence with cleft mandible and limb anomalies | 0.55 | 9.95E-21 | Upregulated |

p-values determined with a Wald test.
FC, fold change; ND, none described.

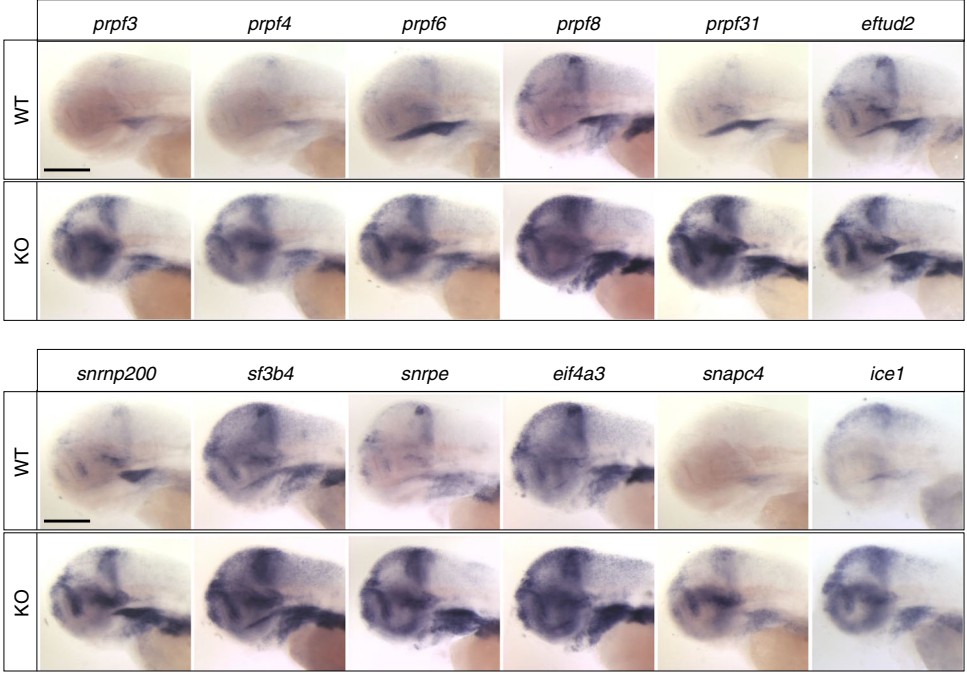

**Fig. 5 Major spliceosome effectors are upregulated in _fam50a_ KO zebrafish.** Representative lateral images of whole-mount in situ hybridization performed on 3 day post-fertilization larvae validate RNA-seq data (Table 2) from _fam50a_ KO (homozygous mutant) vs wild-type (WT). Number of embryos assessed with similar results: _prpf3_, n = 11; _prpf4_, n = 10; _prpf6_, n = 11; _prpf8_, n = 10; _prpf31_, n = 11; _eftud2_, n = 10; _snrnp200_, n = 11; _sf3b4_, n = 11; _snrpe_, n = 11; _eif4a3_, n = 9; _snapc4_, n = 11; _ice1_, n = 10. Scale bars; 200 µm.

spliceosome C-complex[7], we performed a FAM50A pulldown assay followed by mass spectrometry (nanoscale liquid chromatography with tandem MS). We transfected HEK-293T cells with a C-terminally tagged dual streptavidin and flag epitopes on a FAM50A PiggyBac transposon system and expanded cells under puromycin selection. Strepavidin capture identified a combined repertoire of 99 FAM50A interactors with a SAINT probability cutoff value >0.7 (Supplementary Data 1) compared to controls in two independent replicates. The top 20 functional groups involve RNA processing, with U5 snRNA binding (GO:0030623) ranked with the most significant fold-enrichment ($p = 1.3E{-}02$; Kolmogorov–Smirnov test, Supplementary Table 7).

To show direct interaction of FAM50A with bona fide spliceosome effectors, we performed co-immunoprecipitation (co-IP) assays. We transiently transfected U-87 glioblastoma cells with V5-tagged EFTUD2 and DDX41 plasmids, which are part of the spliceosome U5 and C-complex, respectively[7] (Fig. 6a, b). Immunoblotting using anti-V5 and anti-GAPDH antibodies detected the overexpressed tagged proteins in input lysates (Fig. 6c, d). Next, we immunoprecipitated proteins with anti-FAM50A antibody in transfected and negative control samples. Western blot against FAM50A in co-IP lysates discovered the pulled down FAM50A protein in all samples. Immunoblot with anti-V5 in IP lysates detected both Co-IP partners in their respective transfected samples but not in negative controls, indicating a specific physical interaction of FAM50A with spliceosome binding partners that are active during the two-step splicing reaction (Fig. 6c, d).

## Discussion

We report partial loss-of-function missense variants in _FAM50A_ as the genetic basis of Armfield XLID syndrome. Our work epitomizes the challenges of understanding rare disease pathogenesis. Using candidate gene sequencing, we identified the causal _FAM50A_ variant in the original Armfield XLID syndrome family in 2001. Next-generation sequencing technology and data sharing

platforms were required to identify four additional cases with _FAM50A_ variants several years later. Even with bolstered support for genetic causality, this work required a vertebrate model to gain insight into variant pathogenicity and cellular mechanism. Our experience is not unique. Of the estimated 9000 Mendelian phenotypes that have been described[32], a substantial proportion of gene-phenotype pairs identified in the last 5 years required partnering of a rare human finding with a model organism or relevant in vitro functional assay[33].

XLID conditions range from isolated ID to multi-organ disorders. Males with _FAM50A_ variants overlap phenotypically with other XLID syndromes presenting with seizures and dysmorphic facial features (Table 1). We noted variable head circumference with an inconsistent facial phenotype. However, Armfield XLID syndrome also includes postnatal growth retardation and ocular findings, which are less common among XLID syndromes[3,34]. Affected males with _FAM50A_ variants had normal intrauterine growth but statural growth slowed postnatally, eventually falling more than 2 SD below the mean for height. Small hands and feet are also signature features of this cohort. Ocular anomalies were pervasive with all patients having strabismus, keratoconus, and anterior chamber anomalies. In sum, this constellation of cognitive, craniofacial, growth, and ocular features define Armfield XLID syndrome as a distinct clinical entity.

To investigate FAM50A function, we generated zebrafish KO models, however, _fam50a_ KO zebrafish larvae are not viable long term. Homozygous mutants for two independent loss-of-function alleles exhibited defects impacting neuronal proliferation and craniofacial patterning, which mirror the phenotypes in _FAM50A_ mutation-bearing males (Figs. 1 and 2; Table 1). However, there is a key difference between the KO and Armfield XLID syndrome cases: strength of allele effect. Although we were limited to testing variant pathogenicity using either a quantitative comorbid feature (craniofacial patterning) or a clearly defined qualitative morphological defect (swim bladder), data generated through in vivo complementation assays suggested that males with _FAM50A_

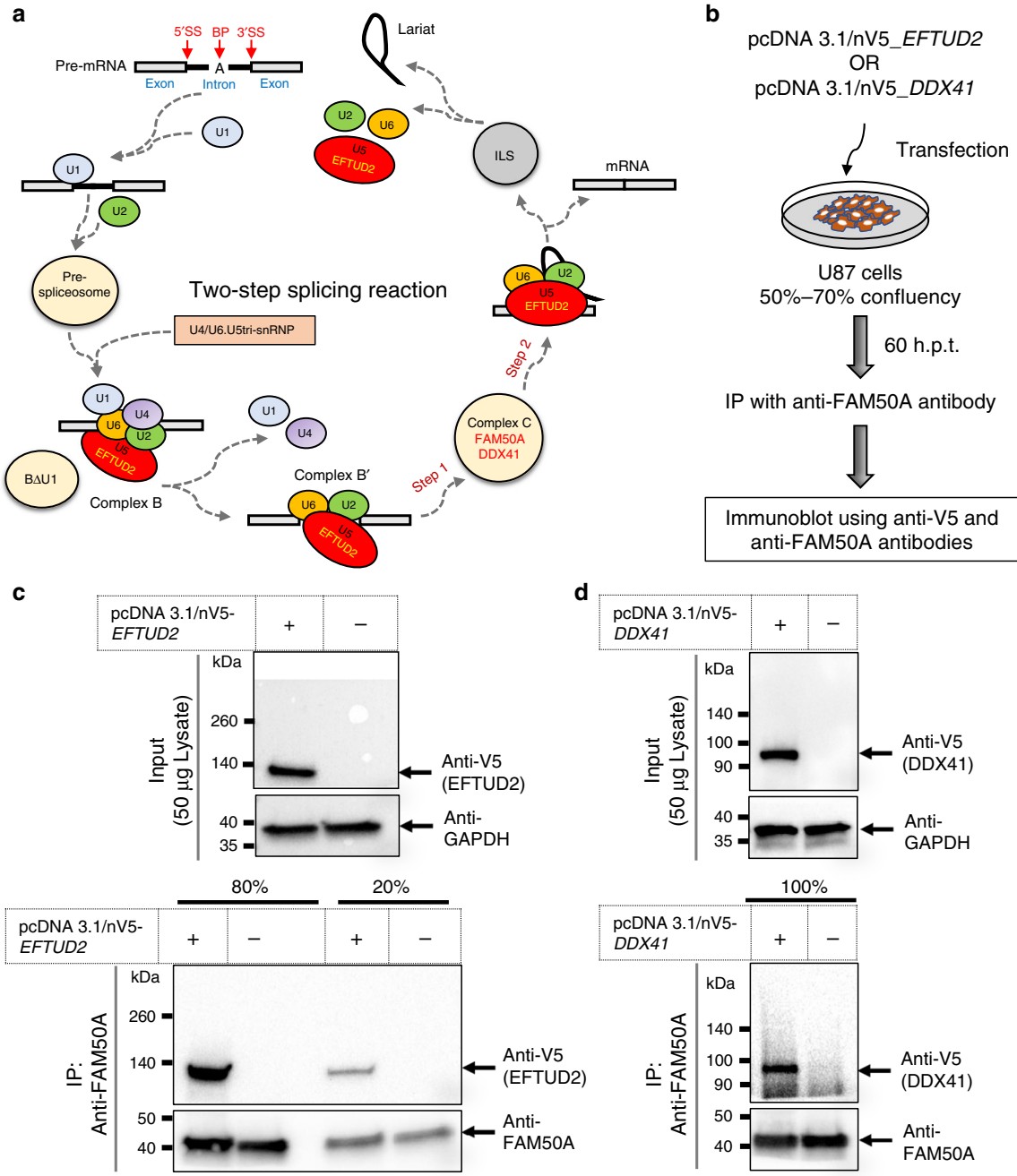

**Fig. 6 FAM50A interacts with U5 and C-complex proteins. a** Schematic representation of the two-step splicing reaction (adapted from ref. [70]). EFTUD2 (U5); FAM50A and DDX41 (C-complex). **b** Graphical representation of semi-native co-immunoprecipitation assay. Candidate interactors were overexpressed in U-87 cells, harvested in immunoprecipitation (IP) lysis buffer at 60 h post-transfection and immunoprecipitated with anti-FAM50A antibody. The interaction partners were detected in IP lysate using anti-V5 and anti-FAM50A antibodies, respectively. **c, d** Western blot of proteins after co-immunoprecipitation. Top: total protein lysate input (50 μg/lane) was migrated on 4–15% polyacrylamide gels to detect EFTUD2 (**c**) or DDX41 (**d**) using anti-V5 mouse monoclonal antibody. GAPDH was used as loading control. Bottom: Anti-FAM50A antibody was used to immunoprecipitate native FAM50A protein in total input lysate of 2.3 mg/condition (**c**) or 3.5 mg/condition (**d**). The IP lysate was separated in two parts (20% and 80%) and migrated independently on 4–15% polyacrylamide gels. The proteins of interest with predicated band sizes are indicated with black arrows. Plus and minus signs indicate presence and absence of relevant plasmids, respectively. EFTUD2 and DDX41 were detected in independent experiments using protein lysates derived from replicate batches of U-87 cells.

variants have some residual protein function (Fig. 3; Supplementary Fig. 13). Even without variant testing data from a direct neurodevelopmental phenotype, we would still expect humans with null *FAM50A* variants to be embryonic lethal, as supported by a low observed/expected (o/e) constraint metric in >125,000 exomes in gnomAD (o/e = 0.06).

mRNA splicing is a dynamic process that involves intron branching and exon ligation[30] (Fig. 6a). Electrophoretic separation of the spliceosome initially identified complexes associated with each splicing step[35,36]. Proteomics and cryo-electron microscopy have refined further this process enabling the contents and relative ratios of spliceosome complexes to be

tabulated[30]. Mass spectrometry of purified active step 1 spliceosomes cataloged the B- and C-complex protein repertoires, and FAM50A was reported as a putative C-complex protein[7,37–41]. In *fam50a* KO zebrafish, GSEA revealed upregulation of functional groups of genes involved in mRNA splicing. rMATS showed significantly altered splice junctions in KO transcriptomes, with an enrichment of events that occur at 3′ splice sites. Transcriptomic profiles obtained from patient-derived LCLs support these data. FAM50A pulldown coupled with mass spectrometry also indicated an enrichment of binding proteins involved in RNA processing and co-IP assays showed direct interaction of FAM50A with U5 and C-complex proteins. These observations are consistent with spliceosome C-complex impairment. However, the specific role of FAM50A and the process by which it is recruited to the spliceosome remains to be elucidated.

Approximately 150 proteins are present in affinity-purified spliceosome C-complex preps from human cells[7]. Other ID syndromes are caused by variants in C-complex-affiliated proteins. *EFTUD2* (B- and C-complex) mutations cause Guion-Almeida-type mandibulofacial dysostosis[22], and animal models display mRNA splicing signatures reminiscent of *fam50a* KO zebrafish[31]. Richieri–Costa–Pereira syndrome (RCPS), caused by variants in *EIF4A3* (B- and C-complex), is a rare autosomal recessive ID syndrome[42]. Further, variants in *THOC2* (B- and C-complex) result in an X-linked recessive condition in which affected males have ID, short stature, seizures, and abnormal gait[43]. Further work will be required to understand whether variants in each C (and B) complex proteins produce similarly aberrant mRNA splicing. Still, the shared phenotypes resulting from complex C protein dysfunction highlights an emergent subclass of spliceosomopathies associated with neurodevelopment.

## Methods

**Human subjects and ethics approval.** All participants in this study were assessed clinically by local physicians with medical genetics expertise. Targeted sequencing and whole-exome sequencing were approved by the Institutional Review Board at each participating center (Greenwood Genetic Center; A.I. duPont Hospital for Children; McMaster University Medical Center; Phoenix Children's Medical Group; University of North Carolina School of Medicine). We obtained signed informed consent for study procedures, publication of genetic findings, and publication of identifiable information (facial photographs) from all participants or their legal representatives. The authors affirm that human research participants provided informed consent for publication of the images in Fig. 1.

**Sequencing and analysis to identify *FAM50A* variants.** For family K8100, the *FAM50A* variant was identified as part of three independent studies: (1) a screen of families linked to Xq28 using exon-specific primers and Sanger sequencing, performed[44]; (2) a next-generation sequencing project of 718 genes located on the X-chromosome[9]; and (3) X-exome next-generation sequencing[45]. For families K9648, K9656, and K9677, trio-based exome sequencing (ES) was conducted by GeneDx.

Using genomic DNA, the exonic regions and flanking splice junctions of the genome were captured using the Clinical Research Exome kit (Agilent Technologies, Santa Clara, CA) or the IDT xGen Exome Research Panel v1.0. Massively parallel (NextGen) sequencing was done on an Illumina system with 100 bp or greater paired-end reads. Reads were aligned to human genome build GRCh37/UCSC hg19, and analyzed for sequence variants using a custom-developed analysis tool and standard protocols were followed for variant interpretation[46]. The general assertion criteria for variant classification are publicly available on the GeneDx ClinVar submission page (http://www.ncbi.nlm.nih.gov/clinvar/submitters/26957/).

For family K9667, WES was done in a research laboratory on the proband, mother and father. Exome capture and library preparation was performed using 2 μg of genomic DNA and the TruSeq DNA sample preparation kit v2 and the TruSeq Exome Enrichment kit v2 (Illumina, San Diego, CA) following manufacturer's guidelines. Sequencing was performed on a HiSeq2000 using multiplexed paired-end sequencing chemistry for 101 bp read length (Illumina, San Diego, CA). Binary base calls in the form of.bcl files were generated by the Illumina HiSeq2000 RTA module during sequencing and were converted to compressed fastq files separated for each index using CASAVA 1.8.2 (Illumina, San Diego, CA). Quality filtered fastq files were aligned to NCBI reference GRCh37.62 with BWA 0.6.2-r126. Binary alignment files were converted and coordinate sorted into the standard BAM format using samtools 0.1.18. Aligned reads were realigned around short insertion and deletions and duplicate reads were filtered using Picard 1.79

(http://broadinstitute.github.io/picard/). This was followed by base quality recalibration with GATK 2.2. Variant calling was done by UnifiedGenotyper and genotype quality recalibrated using VariantRecalibrator and following standard pipelines adopted from best practice methods of GATK 2.2.

In all genetic studies, candidate causal variants were considered if they: (1) affect coding sequence or splice junctions; (2) were absent from healthy control males; and (3) segregated with disease in pedigrees. *FAM50A* variants were confirmed using Sanger sequencing of an independent genomic DNA sample from the proband (Supplementary Table 3). Segregation analysis was conducted on DNA samples from all available family members.

**Determination of X-inactivation.** To determine the X-inactivation status in females in family K8100, we examined the methylation status of the AR locus[11]. We digested ~50–150 ng DNA for 16 h with either *Hpa*II and *Rsa*I or *Rsa*I alone as a control (duplicate samples). Enzymes were then inactivated at 80 °C for 20 min, and the resulting digest was PCR amplified (using one-tenth of the digest as template). We separated amplification products on an ABI3100 and analyzed data with GeneScan software (Applied Biosystems). Raw peak height values obtained from digested samples were normalized for amplification efficiency with mean values from the two *Rsa*I-digested samples to obtain the X-inactivation pattern.

**Structural modeling of the FAM50A protein.** Due to lack of an experimentally-determined structure of FAM50A protein, we used in silico modeling. Homology modeling is not suitable for predicting FAM50A 3D protein structure, since no template can be found that covers the entire sequence of FAM50A. Thus, we used I-TASSER, which is an iterative threading method[47]. We uploaded the full sequence of FAM50A (NP_004690.1) and specified one template structure (PDB: 3AG7), which has 27% sequence identity and covers residues from 173 to 248. In addition, we predicted the secondary structure elements (SSEs) of FAM50 using YASPIN[48].

**Prediction of FAM50A folding free energy change.** The model for FAM50A protein was used to predict folding free energy changes caused by *FAM50A* mutations; we used the following online tools: mCSM[49], SDM[50], DUET[51], SAAF EC[52], and FOLDX[53].

***FAM50A* tissue expression analysis.** The Human Fetal MTC panel (Clontech) was utilized to analyze *FAM50A* expression in fetal tissues. The MTC cDNA preparations were used as template for RT–PCR as outlined by the manufacturer in a final reaction volume of 20 μl (see Supplementary Table 3 for primer sequences). PCR conditions for *FAM50A* were: initial denature 95 °C for 5 min, followed by 30 cycles of: denature at 95 °C, 30 s; annealing at 57 °C, 30 s; and extension at 72 °C, 40 s. Final extension was at 72° for 5 min. For β-actin, the PCR conditions were the same except the annealing was done at 55 °C, the extension was for 30 s and only 25 cycles were used. We migrated 20 μl of the PCR reaction on a 3% agarose gel in 1×TBE (see Source Data for uncropped images).

**Localization of endogenous FAM50A protein in NIH/3T3 cells.** NIH/3T3 cells (American Type Culture Collection, ATCC CRL-1658) were cultured under standard conditions in DMEM supplemented with 10% fetal bovine serum (FBS). Cells were fixed with 4% paraformaldehyde (PFA) at room temperature for 10 min. After washing in PBST (PBS with 0.1% Tween-20), cells were subjected to permeabilization with PBS containing 0.2% Triton X-100 for 10 min and then blocking with 3% BSA in PBS for 40 min. Primary antibody, anti-human FAM50A antibody (1:200; Novus Biologicals, NBP1-89344), which has cross-reactivity to mouse Fam50a protein[54], was treated in PBS containing 0.1% Triton X-100 at 4 °C overnight. Cells were then washed in PBST and incubated with Alexa Fluor 488-conjugated goat anti-rabbit secondary antibody (1:1000; Invitrogen, A-11008) and Hoechst 33342 (1:10,000; Invitrogen, H3570) at room temperature for 1 h. After washing with PBST, the cells were subjected to confocal imaging by using a Nikon A1R confocal microscope (Nikon Instruments) equipped with a Nikon CFI Plan Apochromat VC objective (×60/1.4 numerical aperture (NA); Nikon Instruments) and digital zooming of Nikon imaging software (NIS-element AR 64-bit version 3.21; Laboratory Imaging).

**FAM50A immunoblotting in human LCLs and zebrafish.** Lymphoblast cell lines were obtained by immortalization of lymphocytes from patient blood samples using Epstein-Barr virus. The lymphoblast cell lines were harvested in RPMI-1640 (cat# R8758, Sigma) with 75 ml fetal bovine serum (Atlanta Biological—Lawrenceville, GA, USA) and 5 ml L-Glutamine and 5 ml antibiotic/antimycotic (Sigma-Aldrich—St.Louis, MO) according to standard procedures (K9648, p. Trp206Gly; K9656, p.Glu254Gly and a healthy male control). For total protein extraction, we harvested cells in RIPA lysis buffer containing 50 mM HEPES (pH 7.6), 1% Triton X-100, 0.1% SDS, 50 mM NaCl, 0.5% sodium deoxycholate, 1 mM PMSF (Millipore Sigma), 1× Halt phosphatase inhibitor cocktail (Thermo-Scientific), and 1× complete protease inhibitor cocktail (Millipore Sigma). After protein quantification with BCA protein assay kit (ThermoScientific), we loaded 23 μg heat denatured total protein lysate/lane supplemented with laemmli sample

buffer and reducing agent (NuPAGE™, cat# NP0004; 1×) and migrated on 4–15% SDS-polyacrylamide gel. The protein bands were transferred to Polyvinylidene fluoride (PVDF) membrane and incubated overnight in primary antibody solution (3.5% milk) containing rabbit anti-FAM50A antibody (1:900, Novus Biologicals; NBP1-89344) and mouse anti-GAPDH antibody (1:3000, Santa Cruz Biotechnology, sc-47724). The membrane was then incubated in secondary antibody solution (5% milk in PBST) containing anti-rabbit IgG-HRP (1:4000, Santa Cruz Biotechnology, sc-2357) and anti-mouse IgGκ-HRP (1:4000, Santa Cruz Biotechnology, sc-516102) to detect FAM50A and GAPDH, respectively. We developed the immunoblots using SuperSignal West Pico PLUS Chemiluminescent Substrate (ThermoScientific) per manufacturer's protocol using a ChemiDox XRS imaging system (Bio-Rad). See Source Data for uncropped images. Signal was quantified with Image Studio Lite, and statistical differences were calculated using a Student's t-test.

To test the endogenous protein level in fam50a KO zebrafish, we crossed $fam50^{+/-}$ mutants and harvested larval heads at 2 dpf in complete lysis media. We used tails for genotyping, cross-matched with heads, and 20 heads were pooled per genotype for protein extraction and quantification. 8 µg protein lysate/lane was loaded on 4–15% SDS PAGE and stained with rabbit anti-FAM50A antibody (1:900, Novus Biologicals; NBP1-89344).

**Localization of tagged WT and mutant FAM50A in COS-7 cells**. Total RNA was extracted from lymphoblast cell lines using GenElute Mammalian Total RNA Miniprep (Sigma cat# RTN-70). cDNA was prepared with Superscript First Strand Synthesis Kit for RT–PCR (Invitrogen cat# 11904-018) from 2 µg of RNA prepared from lymphoblast cell lines. A PCR reaction with the oligos (Supplementary Table 3) and PFU Turbo (Stratagene cat# 600250) was employed to generate the insert. The fragment was run on a 1% TAE agarose gel and purified with a Gel Extraction Kit (Qiagen cat# 287040). The purified product was cloned into pcDNA3.1D/V5-His-Topo vector using the pcDNA3.1 Directional Topo Expression Kit (Invitrogen cat# K4900-01). All plasmids generated had a V5 tag on the C terminus. Plasmids were sequenced to confirm the insert (Supplementary Table 3). We used site-directed mutagenesis to introduce alterations (QuikChange II Site-Directed mutagenesis kit, Stratagene catalog number 200524). All constructs were sequence confirmed to verify the mutation and integrity of the ORF.

COS-7 cells were obtained from American Tissue Culture Collection (ATCC CRL-1651), and cultured in DMEM (Sigma cat# D5796) supplemented with 10% FBS (Atlanta Biologicals cat# S12450H), 1× Penicillin/Streptomycin (Sigma cat# P0781), 2 mM glutamine (Sigma catalog number G7513) in a 5% $CO_2$ humidified 37 °C incubator. COS-7 cells were cultured on poly-L-lysine (Sigma cat# P4707) coated 24-well tissue culture dishes in growth media 18–24 h priors to transfection. A transfection complex containing DMEM (Sigma cat# D5796), 1 µg of plasmid, and 2 µl of Lipofectamine 2000 (Invitrogen cat# 11668-027) was prepared and added to each well. After 5 h, the transfection complexes were removed, the cells washed one time with PBS and growth media was added back to the cells. Twenty-four hours post-transfection, the cells were moved to a poly-L-lysine coated glass slip and cultured in growth media for 24 h.

Cells were washed twice with PBS, fixed with 4% PFA/PBS, permeabilized with 0.1% Triton X-100/PBS, and blocked in blocking solution (2% horse sera, 0.4% BSA in PBS). The cells were incubated with anti-V5 mouse monoclonal antibody (1:500, Invitrogen, R960-25), washed with blocking solution and incubated with Alexa Fluor 594 anti-Mouse IgG (1:1000, Invitrogen, A32742) and Alexa Fluor 488 Phalloidin (1:40, Invitrogen, A12379), washed with blocking solution, incubated with DAPI, and mounted with Gold Prolong antifade medium (Invitrogen, P36930). Images were acquired using a Confocal microscope at 60×.

**Zebrafish husbandry**. Animal experiments were conducted in accordance with protocols approved by the Institutional Animal Care and Use Committees (IACUC) at Duke University and Northwestern University, or the Animal Ethics Committee of Chungnam National University (CNU-00866). Adult fish were reared under standard conditions with a 14 h/10 h light/dark cycle. We obtained embryos by natural mating of either WT, transgenic tg(huc:egfp)[55], tg(kdrl:egfp)[56], −1.4col1a1:egfp[57], or mutant tp53[58] adult zebrafish, and embryos were reared in egg water at 28.5 °C. WT and KO zebrafish were obtained from the Zebrafish Center for Disease Modeling (https://cc.aris.re.kr/zcdm).

**Generation of fam50a KO zebrafish**. To identify the FAM50A zebrafish ortholog, we performed a reciprocal BLAST of human FAM50A protein (Ensembl ID: ENST00000393600.7) against the zebrafish genome, and identified a single reciprocal ortholog with 93% similarity and 86% identity (ENSDART00000037501.8). To understand the in vivo role of FAM50A, we established knockout zebrafish model utilizing the CRISPR/Cas9 system[59]. fam50a target sites of CRISPR single guide (sg) RNA and Cas9 were identified using the Optimized CRISPR Design (http://crispr.mit.edu/) and selected oligonucleotides (Supplementary Table 3) were cloned into pDR274 (Addgene) linearized with BsaI (BioLabs). In vitro transcription was carried out using 150–200 ng template and the MaxiScript T7 Kit (Ambion). RNA was precipitated with isopropanol. Cas9 expression vector (from Addgene) was linearized with Dra I (Takara) and purified with an agarose gel DNA extraction kit (ELPIS). Cas9 mRNA was transcribed with the mMESSAGE

mMACHINE®T7 Kit (Ambion) poly (A) tailed with E. coli Poly (A) Polymerase (NEB) and then purified by lithium chloride precipitation following the manufacturer's protocol. One-cell stage zebrafish embryos were injected with 300 ng/µl Cas9 mRNA and 150 ng/µl sgRNA. For genotyping F0, PCR products (20 µl) were re-annealed in a thermal cycler under the following conditions: 95 °C for 2 min, 95 °C to 85 °C at 2 °C/s, 85 °C to 25 °C at 0.1 °C/s, then kept at 4 °C. 16 µl of the re-annealed mixture was incubated with 0.2 µl of T7 endonuclease I, 2 µl of NEB buffer 2 and 1.8 µl of NFW (Nuclease-free water) at 37 °C for 40 min. See Supplementary Table 3 for genotyping primers.

**Transient suppression of fam50a and embryo microinjections**. We designed a splice-blocking morpholino (MO; Supplementary Table 3) targeting the splice donor site of fam50a exon 4. Zebrafish embryos (−1.4col1a1:egfp) were injected with 9 ng MO at the 1–4 cell stage (1 nl/embryo). To determine MO efficiency, embryos were harvested at 2 dpf in Trizol (Invitrogen) and total RNA was extracted following the manufacturer's protocol. RNA was reverse transcribed using the QuantiTect Reverse Transcription kit (QIAGEN), and subsequent RT–PCR was performed by amplifying the region flanking the MO target site (Supplementary Table 3). The resulting product was migrated on a 1% agarose gel and PCR fragments were excised using the QIAquick gel extraction kit (QIAGEN) and cloned into the TOPO TA vector (ThermoFisher). Individual colonies were subjected to bidirectional Sanger sequencing on an ABI 3730 sequencer using BigDye Terminator chemistry (Applied Biosystems) and sequences were analyzed using DNASTAR (Lasergene software). To establish a dose curve titration, we injected 3, 6, and 9 ng of MO. For transient in vivo complementation assays, we used 5 ng of MO.

**Generation of FAM50A plasmids and in vitro transcription**. We obtained a full-length WT human FAM50A ORF construct (GenBank: NM_004699.3; Thermo-Fisher Scientific; IOH4655) and cloned it into the pCS2+ vector using LR clonase II-mediated recombination (ThermoFisher). To generate mutant constructs, we introduced variants found in human cases along with two negative control variants: p.Ala137Val (1 homozygote, 191 hemizygotes in gnomAD) and p.Glu143Lys (1 homozygote, 13 hemizygotes in gnomAD) by site-directed mutagenesis[17]. We designed primers carrying the mutated nucleotide and used the WT FAM50A ORF construct as a template for PCR amplification (Supplementary Table 3). The amplified product was subjected to DpnI digestion to selectively digest the methylated WT template and mutant colonies were obtained by cloning. WT and mutant constructs were sequence confirmed with Sanger sequencing (Supplementary Table 3). We synthesized capped mRNA using linearized pCS2+ vector as template with the mMessage mMachine Sp6 transcription kit (ThermoFisher). For in vivo complementation assays, 150 pg of mRNAs was injected in the presence or absence of MO. For in vivo complementation using KO embryos, we used 300 pg FAM50A mRNA.

**In situ hybridization on whole-mount zebrafish larvae**. Whole-mount in situ hybridization (WISH) was performed using probes for cdkn1a, mdm2, aaas, eftud2, gss, huwe1, mettl16, ice1, prpf3, prpf4, snrnp200, prpf6, snapc4, prpf8, snrpe, prpf31, sf3b4, eif4a3, tp53, pcna, her4.1 (Supplementary Table 3)[60]. Staged embryos were fixed overnight in 4% PFA, and then dehydrated in a methanol gradient. Embryos were then rehydrated in phosphate-buffered saline containing 0.1% Tween-20 (PBST). Embryos were permeabilized by proteinase K digestion and then hybridized with digoxin-labeled probes overnight at 70 °C. The next day, embryos were washed in a preheated mixture of 50% saline sodium citrate containing 0.1% Tween-20 and 50% hybridization solution at 70 °C. Embryos were washed again at room temperature and incubated in staining solution in the dark until sufficient staining appeared. Embryos were mounted in glycerol and were visualized using a Nikon AZ100 microscope (Nikon, Tokyo, Japan). Images were captured using a Nikon DIGITAL SIGHT DS-Fil1 digital camera (Nikon) and processed with NIS-Elements F 3.0 (Nikon).

**Immunostaining on whole-mount zebrafish larvae**. For whole-mount immunofluorescence staining, zebrafish embryos at 24 h post-fertilization (hpf) were fixed overnight in 4% PFA and dehydrated with methanol. Embryos were permeabilized in acetone for 7 min at −20 °C and washed in water, followed by several washes in PBST. After blocking for 30 min in 2% horse serum, zebrafish embryos were incubated with FAM50A antibody (1:200; Novus Biologicals, NBP1-89344) at 4 °C overnight. On the next day embryos were incubated with Alexa Fluor 568-conjugated secondary antibodies (1:500; Life Technologies). For nuclei staining, immunostained zebrafish embryos were counter-stained with Hoechst 33342 at 1 and 10 µg/ml, respectively. For imaging, the yolk was removed and the embryo was flat mounted. Embryo preparations were examined under a confocal microscope (Zeiss LSM700). For confocal imaging, embryos/larvae were mounted in 1.2% low-melting agarose on a glass slide.

**Craniofacial phenotyping in zebrafish larvae**. Cartilage was stained with Alcian Blue[61]. Embryos (2.5, 3 and 4.5 dpf) were fixed in 4% phosphate-buffered PFA overnight. After two washes in PBST for 5 min, embryos were bleached with 6% $H_2O_2$ and 0.5% KOH for 1 h. Following two washes in PBST for 5 min, cartilage

was stained for 3 h in 10 mM $MgCl_2$, 95% EtOH, and 0.04% Alcian blue (A5268, Sigma-Aldrich). After one wash in acidic ethanol (70% ethanol, 5% HCl) followed by an overnight destaining in fresh acidic ethanol, embryos were dehydrated in 85% and 100% ethanol for 15 min each and transferred to 80% glycerol for bright field imaging. To image craniofacial structures in −1.4col1a1:egfp larvae, we performed live automated imaging of the fluorescent signal in mutant or morphant larvae at 3 dpf using the Vertebrate Automated Screening Technology (VAST; Union Biometrica) Bioimager platform. First, larvae were anesthetized with tricaine and passed through a 600 mm borosilicate capillary using VAST default operational settings for 3 dpf larvae. Larvae were rotated to acquire fluorescent ventral images on a Zeiss Axioscope.A1 ×10 (NA 0.3) objective and Axiocam 503 monochromatic camera (ZEN Pro software; Zeiss). Embryos resulting from $fam50a^{+/−}$ matings were subjected one by one for VAST imaging and collected for genotyping after acquisition of live ventral images.

**TUNEL assay on whole-mount zebrafish larvae**. To quantify apoptotic cell death in zebrafish whole mounts, we subjected 2 dpf zebrafish larvae to terminal deoxynucleotidyl transferase dUTP nick end labeling (TUNEL) labeling following the manufacturer's protocol (ApopTag Red In Situ Apoptosis Detection Kit, Millipore Sigma). Larvae were dechorionated at 48 hpf and fixed in PFA overnight at 4 °C followed by 100% methanol fixation at −20 °C overnight. The larvae were then rehydrated and washed with PBST followed by bleaching (0.5% KOH and 3% $H_2O_2$ in PBST) for ~10 min at room temperature. The larvae were then permeabilized using proteinase K (10 μg/ml) and fixed with 4% PFA for 20 min at room temperature. Larvae were then incubated in equilibration buffer (Millipore Sigma) for 1 h at 37 °C and overnight in TdT enzyme (Millipore Sigma) at 37 °C. The next day larvae were washed with PBST, then incubated in anti-Digoxigenin-rhodamine (Millipore Sigma); fluorescent signal was imaged with an AZ100 microscope and Nikon DIGITAL SIGHT DS-Fil1 digital camera.

**Mitotic cell-cycle progression studies in whole-mount larvae**. We used phospho-histone H3 immunostaining to quantify of cell-cycle progression using whole-mount zebrafish larvae fixed at 2 dpf. First, larvae were fixed overnight in PFA at 4 °C followed by 100% methanol fixation overnight at −20 °C. Larvae were then rehydrated with a series of solutions with decreasing percentages of methanol in PBST followed by bleaching for 10 min. The larvae were then permeabilized with proteinase K (10 μg/ml) and post fixed with 4% PFA. Larvae were then incubated in blocking solution (1% FBS, 1% BSA, 0.1% Tween-20 in PBS) for 1 h followed by overnight incubation in anti-phospho-histone 3 (PH3; 1:500, Santa Cruz Biotechnology, sc-8656-R). We performed secondary detection for 1 h (594 anti-rabbit IgG, ThermoFisher; 1:500). After washing with IF buffer (1% BSA, 0.1% Tween-20 in PBS), larvae were imaged with an AZ100 microscope.

**Image processing and statistical analysis of zebrafish data**. Cartilage patterning: We measured the ceratohyal angle using Image J (NIH). Apoptotic and proliferative cell counting: We quantified fluorescently labeled cells in the head (region of interest), using Image J; 8-bit images were inverted and the Image based tool for counting nuclei (ICTN) plugin was used with the following parameters: width: 12, min distance: 6, threshold: 0.5, check detect dark peaks. For all quantitative zebrafish phenotyping assays, the statistical differences were calculated using an unpaired Student's t-test, two-sided (GraphPad Prism).

**Sample preparation for RNA-seq studies**. We performed RNA-seq of five replicates (20 heads per replicate obtained from five independent matings) for each of WT and homozygous $fam50a^{−/−}$ maintained on a WT background. At 2 dpf, we decapitated larvae and subjected tails with proteinase K digestion (Life Technologies) to extract genomic DNA for genotyping via PCR. We harvested the decapitated larval heads in Trizol (Invitrogen). After genotype confirmation, we pooled 20 heads per biological replicate WT and extracted total RNA. For RNA-seq on human LCLs, we isolated total RNA from transformed lymphoblast derived from cases harboring mutations in *FAM50A* (p.Trp206Gly and p.Glu254Gly) and unaffected male control. We then treated with a DNA-free™ DNase treatment kit (ThermoFisher Scientific) per manufacturer's protocol. For RNA quality assurance, we migrated the RNA on screen tape (Agilent Technologies) using a Tape Station 2200 (Agilent Technologies) and the RIN score was determined using Tape Station Analysis Software (Agilent).

**RNA-seq library preparation and sequencing**. Zebrafish RNA-seq libraries were prepared using the KAPA Stranded mRNA-Seq Kit following the manufacturer's protocol. mRNA transcripts were first captured using magnetic oligo-dT beads, fragmented using heat and magnesium, and reverse transcribed using random priming. During the 2nd strand synthesis, the cDNA:RNA hybrid was converted into double-stranded cDNA (dscDNA) and dUTP incorporated into the 2nd cDNA strand. Human RNA-seq libraries were prepared from 1 μg total RNA using the Illumina TruSeq Stranded mRNA Library Preparation Kit according to manufacturer's instructions. Illumina sequencing adapters were ligated to the dscDNA fragments and amplified to produce final RNA-seq libraries. Libraries were indexed and pooled in an equimolar ratio prior to 50 bp single end sequencing on the same

lane of a HiSeq 4000 (Illumina). Sequence data were demultiplexed and Fastq files were generated using Bcl2Fastq conversion software (Illumina).

**RNA-seq data analysis**. Data were processed using the TrimGalore toolkit (http://www.bioinformatics.babraham.ac.uk/projects/trim_galore), which employs Cutadapt[62] to trim low-quality bases and Illumina sequencing adapters from the 3′ end of reads. Only reads that were 20 nt or longer after trimming were kept for further analysis. Reads were mapped to the Zv10r87 version of the zebrafish genome and transcriptome[63] or human genome build hg19 obtained from Ensembl (http://ensembl.org/) using the STAR RNA-seq alignment tool[64] and retained for subsequent analysis if they mapped to a single genomic location. Gene counts were compiled using the HTSeq tool (https://htseq.readthedocs.io/en/master/), and only genes that had ≥10 reads in any given library were used in subsequent analyses. Normalization and differential expression were carried out using the DESeq2[65] Bioconductor[66] package with the R statistical programming environment (https://www.r-project.org/). False discovery rate was calculated to control for multiple hypothesis testing. Gene set enrichment analysis[67] was performed to identify gene ontology terms associated with altered gene expression for each comparison. Alternative splicing was characterized using the rMATS algorithm[29], which employs the STAR alignment tool[64] along with the Zv10r87 version of the zebrafish genome and transcriptome[63] or human genome build hg19 obtained from Ensembl. DAVID pathway analysis[68] was run on significant genes (FDR ≤ 5%) on each of the five alternative splicing event categories as well as the union set of all five lists.

**FAM50A pulldown and mass spectrometry**. A PiggyBac transposon system was used to overexpress C-terminal flag and streptavidin tagged forms of FAM50A protein in HEK-293T/17 cells (ATCC CRL-11268). This coding sequence was cloned into a PiggyBac dual promotor (PB513B-1; System Bioscience) by GeneArt at the multiple cloning site. The tagged construct was expressed under the CMV promoter, puromycin resistance and GFP was expressed with the EF1α promoter. Cells were co-transfected with the engineered PiggyBac dual promoter vector and Super PiggyBac transposase (System Bioscience, PB210PA-1) with Lipofectamine 3000 (Tabletab:PiggyBac-transfection-conditions). 72 h post-transfection, selection was carried out with 2 μg/ml puromycin for 1 week; cells were then expanded prior to proteomic submission.

For unfixed samples, cells were sent as $10^6$ cell aliquots as fresh frozen pellets. For the crosslinking, $10^6$ cell were fixed with 1.5 ml of 1% paraformaldehyde/PBS for 15 min; this was subsequently neutralized with 10 ml 0.125 M glycine solution in PBS, incubated for 5 min, followed by 2× PBS washes before cells were subjected to affinity purification.

For streptavidin based affinity purification, 10–20 × 10e6 293 T cells were lysed in low salt lysis buffer (50 mM Tris-HCl pH 8, 150 mM NaCl, 0.1% Igepal-630, 1 mM EDTA, 1 mM DTT, Halt protease, and phosphatase inhibitors). Whole-cell extracts were incubated with 50 μl of Mag-Strep type 3 XT beads (IBA) for 30 min at 4 °C. Beads with bound proteins were washed 5 times with IPP150 buffer (10 mM Tris-HCl pH 8, 150 mM NaCl, 0.1% Igepal-630) and then three times with 50 mM ammonium bicarbonate. Bound proteins were digested with 0.5 μg of trypsin (sequencing grade, Roche) overnight at 37 °C. The peptide solution was collected, acidified with formic acid to a final concentration of 0.5%, filtered through an HTS Multiscreen filter plate (Millipore) to remove left-over beads and dried in a SpeedVac (Thermo). Peptides were reduced in 50 mM TCEP for 15 min at room temperature and acidified to a final concentration of 0.5% formic acid before mass spectrometry analysis.

Peptides were analyzed by online nanoLC-MS/MS on an Orbitrap Fusion Tribrid mass spectrometer coupled with an Ultimate 3000 RSLCnano System. Samples were first loaded and desalted on a nanotrap (100 μm id × 2 cm) (PepMap C18, 5 μ) at 10 μl/min with 0.1% formic acid for 10 min, and then separated on an analytical column (75 μm id × 50 cm) (PepMap C18, 2 μ) over a 120 min linear gradient of 5–40% B (B = 80% $CH_3CN$/0.1% formic acid) at 300 nl/min, and the total cycle time was 150 min. The Orbitrap Fusion was operated in the Top Speed mode at 3 s per cycle. The survey scans (m/z 375–1500) were acquired in the Orbitrap at a resolution of 120,000 at m/z 200 (AGC 4 × 105 and maximum injection time 50 ms). The multiply charged ions (2–7) with a minimal intensity of 1 × 104 counts were subject to MS/MS in HCD with a collision energy at 30% and an isolation width of 1.6 Th, then detected in the linear ion trap (AGC 1 × 104 and maximum injection time 35 ms). Dynamic exclusion width was set at ±10 ppm for 30 s.

**Mass spectrometry data analysis**. Raw files were processed with Proteome Discoverer v. 1.4 (Thermo). Database searches were performed with Mascot v. 2.2 (Matrix Science) against the human Uniprot database (v. January 2018) appended with the cRAP database (www.thegpm.org/crap/). The search parameters were: trypsin digestion, two missed cleavages, 10 ppm mass tolerance for MS, 0.5 Da mass tolerance for MS/MS, with variable modifications of N-acetylation (protein), oxidation(M), and pyro-glu (N-term Q). Peptide false discovery rates (FDR) were estimated based on matches to reversed sequences in a concatenated target-decoy database using Percolator and set at 0.01. Protein identification required at least one high-confidence peptide (FDR < 1%) with a minimum Mascot score of 20. To

discriminate specific from non-specific interactions, protein lists from bait and control experiments were analyzed with SAINTexpress[69].

**Co-immunoprecipitation (co-IP) using in vitro cell models**. We obtained full-length wild-type open reading frame *EFTUD2* and *DDX41* (ThermoFisher; ID: IOH3606 and ID: IOH11189 respectively) in pENTR221 and cloned it into pcDNA3.1/nV5-DEST Mammalian Expression Vector (ThermoFisher Scientific, cat# 12290010) using LRII clonase-mediated recombination (ThermoFisher). All the vectors were sequence confirmed using bidirectional Sanger sequencing. We maintained U-87 glioblastoma cell lines in MEM Earle's complete media following standard cell culture protocols and transfected cells at 50–70% confluency with 6 μg of plasmid and 12 μl of transfection reagent (X-tremeGENE™ 9 DNA Transfection Reagent; Millipore Sigma; cat# 6365787001).

We performed semi-native Co-IP assays using protein A/G PLUS-Agarose beads (Santa Cruz Biotechnology; cat# sc-2003) following manufacturer's protocol with slight modifications. To extract total protein, we harvested the cells at 60 h post-transfection in IP lysis buffer (Tris. 25 mM; NaCl, 150 mM; EDTA, 1 mM; 1% NP-40 [IGEPAL CA-630]; 5% Glycerol; Phenylmethylsulfonyl fluoride [PMSF; Millipore Sigma; cat# 10837091001-PMSF-RO], 1 mM; 1× complete™ Protease Inhibitor Cocktail [Millipore Sigma; cat# 04693116001]; 1× Halt™ Phosphatase Inhibitor Cocktail [ThermoScientific; cat# 78420]). After protein quantification with BCA protein assay (ThermoScientific; cat# 23225), we used 2.3 mg/ml (EFTUD2) and 3.5 mg/ml (DDX41) of total protein lysate and incubated it with anti-FAM50A antibody (1:111; Novus Biologicals; NBP1-89344) at 4 °C overnight. We added 8 μl of packed gel beads per sample and incubated for 3 h at 4 °C. The beads-antibody conjugate was then washed five times with wash buffer (Tris, 50 mM; NaCl, 300 mM; and Triton X-100, 0.1%) and one time with PBS. The IP lysate was then eluted in elution buffer (SDS, 2%; DTT, 10 mM; Tris, 25 mM; and NaCl, 50 mM) at 95 °C for 5 min. To detect FAM50A and its interaction partners, we used 100% of IP lysate (DDX41) or 20% and 80% of IP lysates (EFTUD2) and migrated protein on a 4–5% SDS-polyacrylamide gel. For transfection efficiency and sample integrity, we used 50 μg of total input protein lysate to detect internal control (anti-GAPDH antibody, 1:4000, Santa Cruz Biotechnology, sc-47724) and overexpressed proteins (anti-V5 antibody, 1:3000, ThermoFisher Scientific; R960-25). See Source Data for uncropped images.

**Reporting summary**. Further information on research design is available in the Nature Research Reporting Summary linked to this article.

## Data availability

Consent restrictions preclude deposition of X-chromosome sequencing data (family K8100) or whole-exome sequencing data (families K9648, K9656, K9667, or K9677). However, specific information (e.g., specific variants, but not full data sets) can be obtained upon request from the corresponding authors. All *FAM50A* variants have been deposited in the ClinVar database under accession numbers VCV000872936.1, VCV000872937.1, VCV000872938.1, VCV000872939.1, and VCV000872940.1. Transcriptomic data were deposited in the NCBI Gene Expression Omnibus under accession numbers GEO: GSE145711 (zebrafish) and GSE145710 (human). Proteomic data were deposited in the ProteomeXchange database under accession number PXD017642. The source data underlying Figs 3b, c, 6c, d and Supplementary Figs 3a, b, 5a, 6d, f, 10a–e, 11b, d, 12, 13b, and 17b, d are provided as a Source data file.

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

## Acknowledgements

We thank the families for their participation, especially members of family K8100 who have been involved in the study of the Armfield syndrome since 1997. Special thanks to R. Nelson who found the *FAM50A* p.Asp255Gly alteration in family K8100 in 1999. We thank Dr. J.T. Koh for sharing the anti-FAM50A antibody. We thank K. Jones for maintaining cell lines and A. Crockett for preparation and editing of the manuscript. We are grateful to N. Devos (Duke Sequencing and Genomic Technologies Core), D. Corcoran (Duke Genomic Analysis and Bioinformatics), X. Wang (Northwestern University Sequencing [NUSeq] Core), and M. Schipma (Northwestern Quantitative Data Science Core), and M. Kedzior for technical assistance. K.K. was funded by an International Research Support Initiative Program fellowship from the Higher Education Commission of Pakistan. This work was supported by grants from the U.S. National Human Genome Research Institute (NHGRI) (T32HG008955 to M.R.M.), National Institute of General Medical Sciences (NIGMS) (R01GM093937 to Y.P. and E.A.), Cancer Research UK (C309/A25144 to J.S.C, M.P, and L.Y.), Korean Ministry of Trade, Industry and Energy (10063396 to C.-H.K.), the National Research Foundation (NRF) of Korea (2018M3A9B8021980 to C.-H.K.), National Institute of Mental Health (NIMH) (R01MH106826 to E.E.D.), National Institute of Neurological Disorders and Storke (NINDS) (R01NS073854 to C.E.S.), and the South Carolina Department of Disabilities and Special Needs (SCDDSN) (2015-45 to C.E.S.).

## Author contributions

Conceptualization, C.-H.K., E.E.D., and C.E.S.; formal analysis, E.A., J.S.C, D.J.A., C.-H.K., E.E.D., and C.E.S.; investigation, Y.-R.L., K.K., S.S., N.A.T, M.P., L.Y., J.W.N., Y.P., T.I.C., S.J.K., H.M.P., D.Y., W.D.H., M.R.M., I.W., A.T., K.M., K.R., M.J.L., and T.M.; resources, K.A.-U., K.G., K.A.A, C.L., C.R., E.S., and C.S.; writing-original draft, K.A.-U., K.G., K.A.A., C.L., E.S., R.E.S., C.-H.K., E.E.D., and C.E.S.; writing-review and editing, Y.-R.L., K.K., N.K., S.M.B., C.-H.K., E.E.D., and C.E.S.; funding acquisition, C.H.K., E.E.D., and C.E.S.

## Competing interests

I.W., A.T., and K.M. are employees of GeneDx, Inc. N.K. is a shareholder in Rescindo Therapeutics. The remaining authors declare no competing interests.
