## [Peer Review File · Nature Communications]

Reviewers' comments:

Reviewer #1 (Remarks to the Author):

This is a thoroughly studied, well described and well written paper reporting 20 years of study leading to the identification of hypomorphic hemizygous variants in FAM50A as disease-causing in Armfield syndrome, an X-linked recessive syndromic form of ID. The authors also decipher the role of Fam50a in splicing regulation via transcriptomic analysis of wt and ko zebrafish larvae. It adds an animal model to tackle the fascinating question of the specificity of phenotypes resulting from spliceosomopathies.

I have only minor comments :

The involvement of FAM50A in the spliceosome complex should be stated in the introduction section.

"recessive variant" should be replaced by hemizygous variant

Selection criteria for likely causal changes are not specified.

line 120 : "to validate the specificity of this finding" do you mean that there are no other candidate variant?

line 120 to 124: 2 sentences giving the same information. delete one

Is there data regarding the inactivation of FAM50A? Is there an inactivation bias in female carriers?

Isn't the right time to change the gene Name?

Reviewer #2 (Remarks to the Author):

The manuscript reports the identification of rare variants in Fam50A, likely causing Armfield XLID syndrome in human. Functional evidence for the gene being involved in this severe syndrome is obtained using a zebrafish genetic KO. The genetic datasets presented are supporting the involvement of the Fam50A in the patient pathologies. However, I would recommend strengthening the functional evidence from the zebrafish model before considering publication further. The conclusion that mutations in FAM50A induce spliceosomopathy is a vast over-interpretation of the current datasets presented in this manuscript. Specific major comments are listed below.

Using the zebrafish mutant to assess the effect of human mutations:

- The human mutations found in patients are predicted to be hypomorphs. Using an animal model losing protein function completely is therefore only indirectly informative. The KO has to be used more directly to assess the phenotype imposed by the patient mutation, by rescuing the null embryos with the human variants. The authors can rescue the mutant by either wildtype or mutant Fam50A mRNA injection (or make a series of UAS:Fam50A transgenics – this latter approach being too long-winded for a rapid publication). Rescue by RNA should be possible as the defects start before day 2 (cell death is detectable at 48hpf already, as shown in Fig. 2D). Moreover, they could also use DNA vectors expressing tagged FAM50A (comparing wt and variants), which will be expressed mosaically and monitor the behaviour of progenitors and neurons in the null. This is a crucial set of experiments needed to understand the impact of the patient isoforms on the CNS.
- The MO experiments described in Fig. 3 are too indirect to properly address the nature of the human mutations. The experiments above need to be done instead.

When is the defect initiated in the zebrafish zygotic nulls:

- As the zebrafish Fam50A is expressed maternally, it is important to know when the protein becomes undetectable in the zygotic null mutants.
- As cell death is dramatically increased in cell populations known to be Wnt-secreting (Fig. 2D) as early as 48hpf, the authors need to follow what is happening to the brain of these mutants

between 24 and 48hpf, looking at HuC and her4 transcripts (the Tg(huc:egfp) line accumulates GFP over time, masking subtle differences). In Fig. 2C, her4 looks weaker in the tectum already at 48hpf.

How specific and direct is the effect on RNAs involved in splicing?:

- Apoptosis has been shown to up-regulate of spliceosome proteins and more generally RNA processing proteins, generating distinct isoforms of apoptotic regulators through alternative splicing. The localisation of these RNAs (Fig.5) to the areas of cell death (Fig. 2D) supports this idea. The authors need to test by PCR whether blocking apoptosis in the mutant reduce the level of upregulation of splicing proteins and the % increase of alternative 3' UTR events.

Reviewer #3 (Remarks to the Author):

Armfield XLID syndrome was previously linked to an 8 Mb region on Xq28. In this study, Lee et al describe FAM50A rare missense variants (D255G, W206G, E254G, R273W, D255N) in 9 affecteds from multiple families with clinical presentations related to Armfield XLID. To clarify the effects of these variants in a vertebrate model, they generate fam50a KO zebrafish models that reproduce some neurodevelopmental and craniofacial patterning defects observed in this XLID syndrome. Complementation assays revealed 5 non-synonymous hypomorphic changes and RNAseq indicated changes in 12% of the transcriptome, with a majority of 3'ss shifts, resulting in loss of RNA transcripts important for brain development. Overall, this is an interesting analysis of additional XLID cases, although the emphasis of this study is on characterization of fam50a KO zebrafish that reproduce some features of Armfield XLID. Additionally, their only experimental readout is abnormal splicing, and the evidence for a direct role of FAM50A in splicing regulation and/or spliceosome function is incomplete.

Major:

1. The progression from discovering FAM50A missense variants that show clinical phenotypes similar to Armfield XLID variants to studying these variants in zebrafish is a reasonable approach. However, the zebrafish KO phenotype is the most striking result so I examined the FAM50A structure and noticed a 5' UTR GCC microsatellite just 5' to the translation initiation codon suggesting the possibility of a pathomechanism similar to FRAXA. Have the authors performed Southern blotting to determine if there is a GCC expansion mutation in Armfield XLID?

2. Results, p7/8 and Fig. S3. Since FAM50A is ubiquitously expressed in fetal and adult tissues, do Armfield XLID patients show a change in the expression level of this gene?

3. Results, p10, ln257. The assertion that the early neurogenesis defects observed in fam50sa KOs, with lethality at ~6dpf in homozygotes, are directly relevant to Armfield XLID is an overstatement.

4. Results, p14, ln353 and Figs. 5 and S13. First, the selection criteria for analyzing these genes was unclear – were these the top 25 genes mis-regulated in the KO? Second, it is interesting that alternative 3'ss changes are the predominant abnormal splicing event, but the authors fail to provide convincing evidence for a direct role of FAM50A in splicing and spliceosomal function and explain why changes in 3'ss selection are primarily affected.

Minor:

5. Abstract. Alterations in splicing leading to alterations in splicing patterns are generally referred to as spliceopathy not spliceosomopathy.

6. Fig. S4A. Using the anti-FAM50A antibody in 3T3 cells, FAM50A protein appears equivalently distributed throughout the nucleus whereas the V5-tagged version in Cos7 cells is concentrated at

the nuclear periphery/NE. The authors should comment on this discrepancy.

7. Fig. S5B. Provide a rationale for CRISPR targeting exons 6,7 to generate KO alleles?

Below are our responses to the reviewers comments.

Reviewer #1:

This is a thoroughly studied, well described and well written paper reporting 20 years of study leading to the identification of hypomorphic hemizygous variants in FAM50A as disease-causing in Armfield syndrome, an X-linked recessive syndromic form of ID. The authors also decipher the role of Fam50a in splicing regulation via transcriptomic analysis of wt and ko zebrafish larvae. It adds an animal model to tackle the fascinating question of the specificity of phenotypes resulting from spliceosomopathies. I have only minor comments:

We thank the reviewer for the positive assessment of our manuscript.

The involvement of FAM50A in the spliceosome complex should be stated in the introduction section.

We are grateful to the reviewer for this suggestion, especially considering the new set of transcriptomics and proteomics data included in the revision. We have added more narrative to the introduction indicating that FAM50A is involved in the spliceosome complex and highlight how our data are consistent with previous reports.

“recessive variant” should be replaced by hemizygous variant

Thank you for your suggestion. We have replaced “recessive variant” with “hemizygous variant”.

Selection criteria for likely causal changes are not specified.

We apologize for this omission. We have added a sentence to the “Sequencing and analysis” section of the methods to clarify this point.

line 120: “to validate the specificity of this finding” do you mean that there are no other candidate variant?

We appreciate this question, and the reviewer understood correctly. Because the original finding was made in 2000, and only four genes were screened, we included a proband in a large resequencing project of 718 genes on the X (2007), as well as a more comprehensive and exhaustive NGS study of genes on the X (2013). We did not detect any other candidate variants in either of these analyses. We have clarified this in the manuscript.

line 120 to 124: 2 sentences giving the same information. delete one

Thank you for raising this point, and we apologize for the confusion. As mentioned above, two different sequencing studies were conducted at different times. The first was by Sanger sequencing of genes on the X and the second using NGS of X chromosome exons. We have clarified this in the revised manuscript.

Is there data regarding the inactivation of FAM50A? Is there an inactivation bias in female carriers?

We thank the reviewer for this question. Indeed, X inactivation had been evaluated as part of another study (Amos-Landgraf et al. 2006) and no bias was observed. These data are now presented in Figure 1a and discussed in the Results section.

Isn't it the right time to change the gene Name?

We respect the reviewer's thoughtful suggestion to change the gene name, however, we believe that it's too early, and also this amendment should be based on prospective functional and structural characterization of FAM50A.

Reviewer #2:

The manuscript reports the identification of rare variants in Fam50A, likely causing Armfield XLID syndrome in human. Functional evidence for the gene being involved in this severe syndrome is obtained using a zebrafish genetic KO. The genetic datasets presented are supporting the involvement of the Fam50A in the patient pathologies. However, I would recommend strengthening the functional evidence from the zebrafish model before considering publication further. The conclusion that mutations in FAM50A induce spliceosomopathy is a vast over-interpretation of the current datasets presented in this manuscript. Specific major comments are listed below.

We thank the reviewer for the positive critique of genetic and functional data, and for motivating us to bolster our functional evidence with additional datasets. We have: (1) added new data showing rescue of mutant zebrafish phenotype (new Supplementary Fig. 13) ; and (2) support FAM50A's role in inducing a spliceosomopathy with transcriptomic data from patient cell lines (new Supplementary Fig. 19; Supplementary Table 5) and FAM50A protein-protein interaction data (new Fig. 6; Supplementary Tables 6 and 7).

Using the zebrafish mutant to assess the effect of human mutations:

- The human mutations found in patients are predicted to be hypomorphs. Using an animal model losing protein function completely is therefore only indirectly informative. The KO has to be used more directly to assess the phenotype imposed by the patient mutation, by rescuing the null embryos with the human variants. The authors can rescue the mutant be either wildtype or mutant Fam50A mRNA injection (or make a series of UAS:Fam50A transgenics – this latter approach being too long-winded for a rapid publication). Rescue by RNA should be possible as the defects start before day 2 (cell death is detectable at 48hpf already, as shown in Fig. 2D). Moreover, they could also use DNA vectors expressing tagged FAM50A (comparing wt and variants), which will be expressed mosaically and monitor the behaviour of progenitors and neurons in the null. This is a crucial set of experiments needed to understand the impact of the patient isoforms on the CNS. The MO experiments described in Fig. 3 are too indirect to properly address the nature of the human mutations. The experiments above need to be done instead.

We recognize reviewer's concern regarding knockdown experiments, and performed additional rescue experiments using stable *fam50a* KO zebrafish. We injected *fam50a* mutant embryos with human mRNA harboring a subset of missense variants (p.W206G, p.E254G, and p.D255N) and evaluated swim bladder formation as a qualitative scoring criterion. In the KO mutant, a swim bladder did not form. Similar to the morpholino rescue experiments, WT human mRNA restored swim bladder formation in all injected embryos. However, we observed restoration of swim bladder formation in a reduced proportion of animals (50-70%) after KO embryos were injected with the subset of missense variants. These data suggest that each mutation is a

hypomorph. These new data are described in the main text and can be visualized in Supplementary Fig. 13. We agree with the reviewer that a series of UAS:FAM50A transgenic models is beyond the scope of the current work. For the MO experiments, we tested the full allelic series with a patient-relevant phenotypic readout (craniofacial cartilage morphology). The specificity of these data are supported by: (1) the observation that a “negative” control variant (common in the gnomAD database) was indistinguishable from WT; and (2) recapitulation of hypomorphic scores for a subset of three variants in the *fam50a* KO rescue studies. We hope that the addition of complementation studies with the *fam50a* KO will mitigate the reviewer’s concerns.

When is the defect initiated in the zebrafish zygotic nulls:

- As the zebrafish Fam50A is expressed maternally, it is important to know when the protein becomes undetectable in the zygotic null mutants.

We thank the reviewer and agree that this is an important point. We reported in the supplementary data of the initial submission that Fam50a protein is undetectable as early as 24 hpf in KO zebrafish. Please see revised Supplementary Fig. 6 and updates to the main text.

As cell death is dramatically increased in cell populations known to be Wnt-secreting (Fig. 2D) as early as 48hpf, the authors need to follow what is happening to the brain of these mutants between 24 and 48hpf, looking at HuC and her4 transcripts (the Tg(huc:egfp) line accumulates GFP over time, masking subtle differences). In Fig. 2C, her4 looks weaker in the tectum already at 48hpf.

We thank the reviewer for this suggestion. We tried to examine changes in neural markers (*her4* and *huc*) in *fam50a* KO zebrafish. *fam50a* mRNA and FAM50A protein levels were reduced as early as 24 hpf, and apoptosis markers (*tp53*, *mdm2*, and *cdkn1a*) were also increased as early as 48 hpf. However, *her4* and *huc* did not demonstrate any observable changes at 48 hpf. Following to reviewer’s suggestion, we carefully reexamined *her4* expression, especially in the tectum at 48hpf. As result, we could not find significant changes in *her4* expression in *fam50a* KO zebrafish, compared to WT and *fam50a*^{+/-}, although there are slight variations among siblings. However, at 3 dpf (72 hpf), there was a dramatic decrease in *her4* expression in *fam50a* KO zebrafish. We provide it as revised Supplementary Fig. 7.

How specific and direct is the effect on RNAs involved in splicing?:

- Apoptosis has been shown to up-regulate of spliceosome proteins and more generally RNA processing proteins, generating distinct isoforms of apoptotic regulators through alternative splicing. The localisation of these RNAs (Fig.5) to the areas of cell death (Fig. 2D) supports this idea. The authors need to test by PCR whether blocking apoptosis in the mutant reduce the level of upregulation of splicing proteins and the % increase of alternative 3’ UTR events.

Thank you. The reviewer has raised a valid concern. To investigate whether blocking apoptosis could rescue the neuronal and splicing defects in zebrafish, we generated a double KO of *fam50a* and *p53*. The double KO showed similar phenotypes to those of the single *fam50a* KO. These data are consistent with the observation that neural markers (*her4* and *huc* at 72hpf) are changed far later than apoptosis markers (*tp53*, *mdm2*, and *cdkn1a* at 48hpf). By RNA *in situ* hybridization, we also show that the upregulation of splicing proteins in *fam50a* KO zebrafish is not changed in double KO of *fam50a* and *p53*. Thus, we can speculate that the *fam50a* KO phenotype is not dependent on the *p53* apoptosis pathway. We have added new experimental data in the revised manuscript text and new Supplementary Fig. 18.

Reviewer #3:

Armfield XLID syndrome was previously linked to an 8 Mb region on Xq28. In this study, Lee et al describe FAM50A rare missense variants (D255G, W206G, E254G, R273W, D255N) in 9 affecteds from multiple families with clinical presentations related to Armfield XLID. To clarify the effects of these variants in a vertebrate model, they generate fam50a KO zebrafish models that reproduce some neurodevelopmental and craniofacial patterning defects observed in this XLID syndrome. Complementation assays revealed 5 non-synonymous hypomorphic changes and RNAseq indicated changes in 12% of the transcriptome, with a majority of 3'ss shifts, resulting in loss of RNA transcripts important for brain development. Overall, this is an interesting analysis of additional XLID cases, although the emphasis of this study is on characterization of fam50a KO zebrafish that reproduce some features of Armfield XLID. Additionally, their only experimental readout is abnormal splicing, and the evidence for a direct role of FAM50A in splicing regulation and/or spliceosome function is incomplete.

Thank you for your positive critique. We agree with reviewer's concern regarding our interpretation of FAM50A's direct role in mRNA splicing, and we conducted further studies to support our claims. As mentioned in responses to Reviewers 1 and 2, these data include: (1) transcriptomic data from FAM50A patient lymphocyte cells; (2) proteomics data generated from FAM50A pulldown experiments; and (3) co-immunoprecipitation studies validating direct interaction between FAM50A and *bona fide* spliceosome components EFTUD2 (U5 component) and DDX41 (C complex component).

Major:

1. The progression from discovering FAM50A missense variants that show clinical phenotypes similar to Armfield XLID variants to studying these variants in zebrafish is a reasonable approach. However, the zebrafish KO phenotype is the most striking result so I examined the FAM50A structure and noticed a 5' UTR GCC microsatellite just 5' to the translation initiation codon suggesting the possibility of a pathomechanism similar to FRAXA. Have the authors performed Southern blotting to determine if there is a GCC expansion mutation in Armfield XLID?

We appreciate the reviewer noting the presence of a GGC microsatellite in the 5' UTR of *FAM50A*. Indeed, an expansion was ruled out before proceeding to sequencing of the candidate genes. This point is clarified further in the paper.

2. Results, p7/8 and Fig. S3. Since FAM50A is ubiquitously expressed in fetal and adult tissues, do Armfield XLID patients show a change in the expression level of this gene?

We thank the reviewer for raising this important question. We investigated FAM50A protein abundance in lymphoblast cell lines derived from two FAM50A cases and a male control individual. Immunoblotting with anti-FAM50A antibody revealed no detectable differences relative to control. These new data are included in the main text of the manuscript and new Supplementary Fig. 5.

3. Results, p10, ln257. The assertion that the early neurogenesis defects observed in fam50sa KOs, with lethality at ~6dpf in homozygotes, are directly relevant to Armfield XLID is an overstatement.

We appreciate the reviewer's concern about "overstatement". It is likely that a complete null mutation causes lethality in humans since we failed to generate null KO mice at three separate institutions. We thus conclude that human patients may survive due to hypomorphic variants. Given the complex neurological phenotypes in humans with Armfield syndrome (intellectual disability, seizures, ventricular phenotypes, hypotonia, tremor, and corpus callosum defects), the most likely explanation is that the neurogenesis defects in *fam50a* zebrafish are relevant to the human phenotype. We submit, however, that further work will be required to understand fully the cellular basis for the human phenotypes.

4. Results, p14, In353 and Figs. 5 and S13. First, the selection criteria for analyzing these genes was unclear – were these the top 25 genes mis-regulated in the KO?

We thank the reviewer for raising this point. Eight genes (*SNAPC4*, *ICE1*, *TP53*, *MDM2*, *METTL16*, *PRPF3*, *PRPF31*, *EFTUD2*) are among the top ten transcripts with the greatest fold change in upregulation. The rest of the dysregulated genes were randomly selected for validation of RNA-seq data; these included another seven mRNA splicing effector transcripts; five significantly depleted transcripts; and genes assessed by WISH during initial characterization of the *fam50a* KO. We have clarified this point in the text.

Second, it is interesting that alternative 3'ss changes are the predominant abnormal splicing event, but the authors fail to provide convincing evidence for a direct role of FAM50A in splicing and spliceosomal function and explain why changes in 3'ss selection are primarily affected.

We are grateful to the reviewer for raising this concern, because it prompted us to do further work that strengthened the paper. As described above and in responses to Reviewers 1 and 2, these data include: (1) transcriptomic data from Armfield syndrome patient lymphocyte cells; (2) proteomics data generated from FAM50A pulldown experiments; and (3) co-immunoprecipitation studies validating direct interaction between FAM50A and *bona fide* spliceosome components EFTUD2 (U5 component) and DDX41 (C complex). We also hope that the schematic in new Fig. 6a will aid the reviewer in understanding the 3'splice site selection impairment.

Minor:

5. Abstract. Alterations in splicing leading to alterations in splicing patterns are generally referred to as spliceopathy not spliceosomopathy.

Thank you for raising this point. Considering the new set of transcriptomics and proteomics data included in the revision, we hope the reviewer will agree that the spliceosome is now implicated directly in pathology. Respectfully, we would like to maintain the more specific term "spliceosomopathy" throughout the manuscript.

6. Fig. S4A. Using the anti-FAM50A antibody in 3T3 cells, FAM50A protein appears equivalently distributed throughout the nucleus whereas the V5-tagged version in Cos7 cells is concentrated at the nuclear periphery/NE. The authors should comment on this discrepancy.

Thank you. This discrepancy could be due to multiple reasons. First, it could simply be due to the use of different cell types. Also, in case of 3T3 cells we detected the endogenous protein, while in case of Cos7 cells we overexpressed FAM50A. We speculate that artificial

overexpression and exogenous epitope tag may partly affect its localization. We have clarified this in the main text.

7. Fig. S5B. Provide a rationale for CRISPR targeting exons 6,7 to generate KO alleles?

The reviewer raises a good point. In several cases, we experienced that targeting of the first exon can lead to the use of alternative translational start sites and no phenotype in KO zebrafish. For this reason, we recently changed our strategy to generate mutants by targeting the most conserved domain between zebrafish and humans, in order to better provide a complete KO of the gene. We have clarified this logic in the main text.

Reviewer #3 (Remarks to the Author):

The authors have addressed all of my concerns.

Reviewer #4 (Remarks to the Author):

The revisions added to the manuscript have clarified and enhanced the examination of Fam50A. The new data indicating a role of Fam50A in spliceosomopathy is compelling, and has strengthened the paper. Some more experiments need to be done, however, to fully address Reviewer #2's concerns.

Reviewer #2 argues that a zebrafish knockout model is only indirectly informative, and that the knockout needs to be rescued with mutant versions of fam50a mRNA. The authors have performed this rescue, but only use swim bladder formation as a readout of the mutant phenotype. Swim bladder formation is unrelated to any patient-relevant phenotype and is therefore uninformative with regards to the effect of the mutations. The cartilage-patterning defects examined in the morpholino experiment are of interest, but they only indirectly address the neuronal changes that would be expected in an intellectual disability model. To assess the effect of the FAM50A variants, the rescue of the knockout should be assessed by *huc*, *her4.1*, and a cell death marker

Responses to Reviewer Comments

Reviewer #4:

The revisions added to the manuscript have clarified and enhanced the examination of Fam50A. The new data indicating a role of Fam50A in spliceosomopathy is compelling, and has strengthened the paper.

Thank you for your positive feedback!

Some more experiments need to be done, however, to fully address Reviewer #2's concerns. Reviewer #2 argues that a zebrafish knockout model is only indirectly informative, and that the knockout needs to be rescued with mutant versions of *fam50a* mRNA. The authors have performed this rescue, but only use swim bladder formation as a readout of the mutant phenotype. Swim bladder formation is unrelated to any patient-relevant phenotype and is therefore uninformative with regards to the effect of the mutations. The cartilage-patterning defects examined in the morpholino experiment are of interest, but they only indirectly address the neuronal changes that would be expected in an intellectual disability model. To assess the effect of the *FAM50A* variants, the rescue of the knockout should be assessed by *huc*, *her4.1*, and a cell death marker

We acknowledge these points from the reviewer, and are grateful for the suggestion to assess the effect of the *FAM50A* patient variants in neuronal cell types in the *fam50a* KO. We have completed this experiment (see Figure on the next page), which was technically feasible with our existing tools. However, we are unable to overcome the interpretive challenges of scoring variants in a neuronal phenotype for the following reasons:

1. The variants are hypomorphic (partial loss of function). Data from both the transient *in vivo* complementation studies using quantitative cartilage patterning as a phenotypic readout (morpholino plus WT mRNA vs variant mRNA; Figure 3) as well as the swim bladder assay in stable mutants (KO plus WT mRNA vs variant mRNA; Supplementary Figure 13) support this conclusion. Thus, to capture differences between function of WT and variant protein, a robust quantitative readout is required to detect differences.
2. Wholemout RNA ISH is not quantitative. Variant scoring involves comparison between different experimental groups using either clearly defined qualitative criteria (as shown in the swim bladder assay; presence/absence), or with a quantitative trait (e.g. ceratohyal angle). Whereas RNA ISH was informative to show developmental defects in *fam50a* KO zebrafish larvae (Figure 2, Supplementary Figures 7 and 8), it is not feasible to either define qualitative scoring criteria or to quantify the intensity of the staining. This is especially poignant for differentiating between changes that we expect to be modest based on our existing data.
3. Alternatives to evaluate neurodevelopment are limited with our current tools. Humans with Armfield XLID syndrome have functional deficits (developmental delay, intellectual disability, and seizures) but no unifying set of structural defects that could be measured on orthologous neuroanatomical structures in the developing zebrafish. This information motivated our initial strategy for testing variants with a co-morbid feature that could be measured (dysmorphic facial gestalt).

In sum, we are not comfortable adding the RNA ISH data to the manuscript due to the interpretive challenges described. We have added text to the manuscript (Pages 20-21) to acknowledge the limitation of not testing variants directly with a neurodevelopment phenotype.

Figure. Representative images of wholemount RNA *in situ* hybridization data generated to assay FAM50A variant function (example: p.D255N). Embryos resulting from *fam50a*^{+/-} x *fam50a*^{+/-} crosses were injected with 300 pg human *FAM50A* mRNA, fixed at 3 days post-fertilization, and assessed with molecular markers of neurogenesis (*her4.1*, *huc*), cell death (*tp53*), or cell proliferation (*pnca*). Lateral (*her4.1*, *huc*, and *tp53*) and dorsal (*pnca*) views are shown. Subtle differences are detectable in the following regions between WT and variant mRNA injected batches: *her4.1*- dorsal aspect of the forebrain (red arrow); *huc*- eye (red dashed circle), and dorsal aspect of the forebrain; *tp53*- mandible; *pnca*- eye (red asterisk), forebrain, and midbrain. Number of embryos used for *her4.1 in situ*: uninjected control, n=12; WT, n=12; D255N, n=16; for *huc*, uninjected control, n=10; WT, n=20; D255N, n=12; for *tp53*, uninjected control, n=10; WT, n=12; D255N, n=17; and for *pnca*, uninjected control, n=16; WT, n=84; D255G, n=108.